# NATURAL ATTRIBUTE-BASED SHIFT DETECTION

## ABSTRACT

Despite the impressive performance of deep networks in vision, language, and healthcare, unpredictable behaviors on samples from the distribution different than the training distribution cause severe problems in deployment. For better reliability of neural-network-based classifiers, we define a new task, natural attribute-based shift (NAtS) detection, to detect the samples shifted from the training distribution by some natural attribute such as age of subjects or brightness of images. Using the natural attributes present in existing datasets, we introduce benchmark datasets in vision, language, and medical domain for NAtS detection. Further, we conduct an extensive evaluation of prior representative out-of-distribution (OOD) detection methods on NAtS datasets and observe an inconsistency in their performance. To understand this, we provide an analysis on the relationship between the location of NAtS samples in the feature space and the performance of distance- and confidence-based OOD detection methods. Based on the analysis, we split NAtS samples into three categories and further suggest a simple modification to the training objective to obtain an improved OOD detection method that is capable of detecting samples from all NAtS categories.

## 1 INTRODUCTION

Deep learning has significantly improved the performance in various domains such as computer vision, natural language processing, and healthcare (Bojarski et al., 2016; Mikolov et al., 2013; Esteva et al., 2016). However, it has been reported that the deep classifiers make unreliable predictions on samples drawn from a different distribution than the training distribution (Amodei et al., 2016; Hendrycks & Gimpel, 2017; Nguyen et al., 2015). Detection of unreliable predictions is important to build a robust model but it is relatively hard when the test distribution is gradually shifting due to a natural attribute since it is difficult to determine whether the classifier fails when the shift is not significant. These shifts occur in the real-world as a result of a change in specific attribute. For example, a clinical text-based diagnosis classifier trained in 2021 will gradually encounter increasingly shifted samples as time flows, since writing styles change and new terms are introduced in time. Detection of such samples is a vital task especially in safety-critical systems, such as autonomous vehicle control or medical diagnosis, where wrong predictions can lead to dire consequences. To this end, we take a step forward by proposing a new task of detecting samples shifted by a natural attribute (*e.g.*, age, time) that can easily be observed in the real-world setting. We refer to such shifts as *Natural Attribute-based Shifts (NAtS)*, and the task of detecting them as NAtS detection.

Detection of NAtS is both different from, and also more challenging than out-of-distribution (OOD) detection (Hendrycks & Gimpel, 2017; Liang et al., 2018; Lee et al., 2018; Chandramouli & Sageev, 2020), which typically evaluates the detection methods with a clearly distinguished in-distribution (ID) samples and OOD samples (*e.g.*, CIFAR10 as ID and SVHN as OOD, which have disjoint labels). In contrast, we aim to detect samples from a natural attribute-based shift within the same label space. Since NAtS samples share more features with the ID than the typical OOD samples do, identifying the former is expected to be more challenging than the latter. Although OOD detection has some relevance to NAtS detection, comprehensive evaluation of the existing OOD detection methods on the natural attribute-based shift is an unexplored territory. Therefore, in this paper, we perform an extensive evaluation of representative OOD methods on NAtS samples.

Depending on the task environment, NAtS detection can be pursued in parallel to domain generalization (Seo et al., 2020; Gulrajani & Lopez-Paz, 2020; Carlucci et al., 2019), which aims to overcome domain shifts (*e.g.*, image classifier adapting to sketches, photos, art paintings, etc.). For example, an X-ray-based diagnosis model should detect images of unusual brightness so that the X-ray ma-

chine can be properly configured, and the diagnosis model can perform in the optimal setting. In other cases, domain generalization can be preferred, such as when we expect the classifier to be deployed in a less controlled environment (*e.g.*, online image classifier) for non-safety critical tasks.

In this paper, we formalize NAtS detection to enhance the reliability of real-world decision systems. Since there exists no standard dataset for this task, we create a new benchmark dataset in the vision, text, and medical domain by adjusting the natural attributes (*e.g.*, age, time, and brightness) of the ID dataset. Then we conduct an extensive evaluation of representative confidence- and distance-based OOD methods on our datasets and observe that none of the methods perform consistently across all NAtS datasets. After a careful analysis on where NAtS samples reside in the feature space and its impact on the distance- and confidence-based OOD detection performance, we identify the root cause of the inconsistent performance. Following this observation, we define three general NAtS categories based on two criteria: the distance between NAtS samples and the decision boundary, the distance between NAtS samples, and the ID data. Finally, we conduct an additional experiment to demonstrate that a simple modification to the negative log-likelihood training objective can dramatically help the Mahalanobis detector (Lee et al., 2018), a distance-based OOD detection method, generalize to all NAtS categories. We also compare our results with various baselines and show that our proposed modification outperforms the baselines and is effective across the three NAtS datasets.

In summary, the contributions of this paper are as follows:

- We define a new task, *Natural Attribute-based Shift detection (NAtS detection)*, which aims to detect the samples from a distribution shifted by some natural attribute. We create a new benchmark dataset and provide them to encourage further research on evaluating NAtS detection.

- To the best of our knowledge, this is the first work to conduct a comprehensive evaluation of the OOD detection methods on shifts based on natural attributes, and discover that none of the OOD methods perform consistently across all NAtS scenarios.

- We provide novel analysis based on the location of shifted samples in the feature space and the performance of existing OOD detection methods. Based on the analysis, we split NAtS samples into three categories.

- We demonstrate that a simple yet effective modification to the training objective for deep classifiers enables consistent OOD detection performances for all NAtS categories.

## 2    NATURAL ATTRIBUTE-BASED SHIFT DETECTION

We now formalize a new task, NAtS detection, which aims to enhance the reliability of real-world decision systems by detecting samples from NAtS. We address this task in the classification problems. Let $\mathcal{D}_I = \{\mathcal{X}, \mathcal{Y}\}$ denote the in-distribution data, which is composed of $N$ training samples with inputs $\mathcal{X} = \{x_1, ..., x_N\}$ and labels $\mathcal{Y} = \{y_1, ..., y_N\}$. Specifically, $x_i \in \mathbb{R}^d$ represents a $d$-dimensional input vector, and $y_i \in \mathcal{K}$ represents its corresponding label where $\mathcal{K} = \{1, ..., K\}$ is a set of class labels. The discriminative model $f_\theta : \mathcal{X} \to \mathcal{Y}$ learns with ID dataset $\mathcal{D}_I$ to assign label $y_i$ for each $x_i$. In the NAtS detection setting, we assume that an in-distribution sample consists of attributes, and some of the attributes can be shifted in the test time due to natural causes such as time, age, or brightness.

When a particular attribute $A$ (*e.g.*, age), which has a value of $a$ (*e.g.*, 16), is shifted by the degree of $\delta$, the shifted distribution can be denoted as $\mathcal{D}_S^{A=a+\delta} = \{\mathcal{X}', \mathcal{Y}'\}$. $\mathcal{X}' = \{x'_1, ..., x'_M\}$ and $\mathcal{Y}' = \{y'_1, ..., y'_M\}$ represents the $M$ shifted samples and labels respectively. Importantly, in the NAtS setting, although the test distribution is changed from the ID, the label space is preserved as $\mathcal{K}$, which is the set of class labels in $\mathcal{D}_I$. In the test time, the model $f_\theta$ might encounter the sample $x'$ from a shifted data $\mathcal{D}_S^{A=a+\delta}$, and it should be able to identify that the attribute-shifted sample is not from the ID.

## 3    NATS DATASET DESCRIPTION

In this section, we describe three benchmark datasets which have a controllable attribute for simulating realistic distribution shifts. Since there exists no standard dataset for NAtS detection, we create new benchmark datasets using existing datasets by adjusting natural attributes in order to reflect real-world scenarios. We carefully select datasets from vision, language, and medical domains containing natural attributes (*e.g.*, year, age, and brightness), which allows us to naturally split the

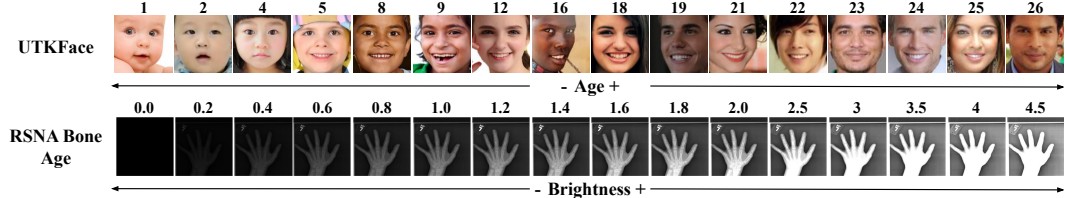

Figure 1: Facial images from the UTKFace dataset to show the variation with age. X-ray images with different levels of brightness created from the RSNA Bone Age dataset.

samples. By grouping samples based on these attributes, we can induce natural attribute-based distribution shifts as described below.

**Image.** We use the UTKFace dataset (Zhang et al., 2017) which consists of over $20,000$ face images with annotations of age, gender, and ethnicity. As shown in Figure 1, we can visually observe that the facial images vary with age. Therefore, we set the $1,282$ facial images of 26 years old age as $\mathcal{D}_{\mathrm{I}}$. For creating the NAtS dataset, we vary the age of UTKFace dataset. To obtain an equal number of samples in the NAtS dataset, the age groups that has less than 200 images are merged into one group until it has 200 samples. Finally, 15 groups $\mathcal{D}_{\mathrm{S}}^{\mathrm{age}}$ are produced for the NAtS datasets, varying the ages from 25 to 1 (*i.e.*, $\mathcal{D}_{\mathrm{S}}^{\mathrm{age}=25}, \mathcal{D}_{\mathrm{S}}^{\mathrm{age}=24}, \ldots, \mathcal{D}_{\mathrm{S}}^{\mathrm{age}=1}$).

**Text.** We use the Amazon Review dataset (He & McAuley, 2016; McAuley et al., 2015) which contains product reviews from Amazon. We consider the product category "book" and group its reviews based on the year to reflect the distributional shift across time. We obtain 9 groups with each group containing reviews from the year between 2005 and 2014. Then, the group with $24,000$ reviews posted in 2005 is set as $\mathcal{D}_{\mathrm{I}}$, and the groups with reviews after 2005 as $\mathcal{D}_{\mathrm{S}}^{\mathrm{year}}$ (*i.e.*, $\mathcal{D}_{\mathrm{S}}^{\mathrm{year}=2006}, \mathcal{D}_{\mathrm{S}}^{\mathrm{year}=2007}, \ldots, \mathcal{D}_{\mathrm{S}}^{\mathrm{year}=2014}$). Each $\mathcal{D}_{\mathrm{S}}^{\mathrm{year}}$ group contains 1500 positive reviews and 1500 negative reviews. We observed that as we move ahead in time, the average length of a review gets shorter and it uses more adjectives than previous years. Due to the space constraint, we provide a detailed analysis of the dataset in the Section B of the Appendix.

**Medical.** We use the RSNA Bone Age dataset (Halabi et al., 2019), a real-world dataset that contains left-hand X-ray images of the patient, along with their gender and age (0 to 20 years). We consider patients in the age group of 10 to 12 years for our dataset. To reflect diverse X-ray imaging set-ups in the hospital, we varied the brightness factor between 0 and 4.5 and form 16 different dataset $\mathcal{D}_{\mathrm{S}}^{\mathrm{brightness}}$ (*i.e.*, $\mathcal{D}_{\mathrm{S}}^{\mathrm{brightness}=0.0}, \mathcal{D}_{\mathrm{S}}^{\mathrm{brightness}=0.2}, \ldots, \mathcal{D}_{\mathrm{S}}^{\mathrm{brightness}=4.5}$), and each group contains X-ray images of 200 males and 200 females. Figure 1 presents X-ray images with different levels of brightness with realistic and continuous distribution shifts. In-distribution data $\mathcal{D}_{\mathrm{I}}$ is composed of $3,000$ images of brightness factor $1.0$ (unmodified images).

## 4 CAN OOD DETECTION METHODS ALSO DETECT NATS?

In this section, we briefly discuss about OOD detection methods and conduct an extensive evaluation of OOD detection methods on our proposed benchmark datasets.

### 4.1 OOD DETECTION METHODS

In this work, we use three widely-used post-hoc and modality-agnostic OOD detection methods. We use **maximum of softmax probability (MSP)** (Hendrycks & Gimpel, 2017) and **ODIN** (Liang et al., 2018) as confidence-based OOD detection baselines, and **Mahalanobis detector** (Lee et al., 2018) as distance-based OOD detection baseline. Note that **ODIN** and **Mahalanobis detector** assume the availability of OOD validation dataset to tune their hyperparameters. However, for all our experiments, we use variants of the above methods that do not access the OOD validation dataset as Hsu et al. (2020). The exact equations and details of how each OOD detection method assigns an OOD score to a given sample is provided in Section A of the Appendix.

### 4.2 EXPERIMENTS AND RESULTS

We now systematically evaluate the performance of the three OOD detection methods under NAtS. We report the AUROC of all OOD detection methods averaged across five random seeds, evaluated for all NAtS datasets. The details of the implementation are described in Section C of the Appendix.

**Experimental Settings.** In image domain, we train a gender classification model on our UTKFace NAtS dataset using ResNet18 model and the cross-entropy loss. We use our Amazon Book Review

Figure 2: Comparison of well-known distance-based and confidence-based OOD detection methods for the CE model on our benchmark datasets. Age '26', year '2005', and brightness '1.0' are in-distribution data in UTKFace, Amazon Review, and RSNA Bone Age dataset, respectively. The NAtS detection performance of these methods is inconsistent across different datasets.

NAtS dataset in text domain and train a 4-layer Transformer with the cross-entropy loss for the sentiment classification task. Lastly, in medical domain, we use our RSNA Bone Age NAtS dataset and train a ResNet18 with cross-entropy loss to predict the gender given the hand X-ray image of the patient. We then evaluate the trained model on the corresponding test set in image, text, medical domains, respectively. Further, we evaluate the NAtS detection performance of representative OOD detection methods in image, text, and medical domains on their corresponding NAtS datasets, which gradually shift with age, year, and brightness, respectively.

**Results.**    We present the classification accuracy of the trained models on the ID test set in Table 1. We observe that the models trained using the cross-entropy loss obtain high accuracy and perform well on their corresponding tasks. We further demonstrate the effectiveness of the existing representative OOD detection methods on our benchmark datasets in Figure 2. The higher AUROC indicates that more NAtS samples are distinguished from the ID samples using OOD detection methods. To provide the reference point, we calculate the AUROC values where there is no shift in the attribute. We observe that in the UTKFace NAtS dataset, samples are detected by ODIN and MSP, which are confidence-based methods, but not by Mahalanobis detector (Figure 2a). In the Amazon Review dataset, NAtS samples are detected only by the Mahalanobis detector, while MSP and ODIN fail (Figure 2b). Moreover, the scores of confidence-based methods are lower than 50 in AUROC. Lastly, Figure 2c shows that inputs from NAtS in the RSNA Bone Age dataset are detected well by all three methods.

## 5    ANALYZING INCONSISTENCY OF OOD DETECTION METHODS

In this section, we first study the behavior of NAtS samples in three datasets using PCA visualization. Then we analyze the inconsistent performance of the OOD detection methods by considering them in two categories, namely confidence-based and distance-based methods. Lastly, based on the analysis, we conclude this section by defining three NAtS categories.

### 5.1    ANALYSIS OF THE LOCATION OF NATS SAMPLES

As illustrated in Figure 3, we apply principal component analysis (PCA) on the feature representations obtained from the penultimate layer of the models to visualize the movement of NAtS samples as we monotonically increase the degree of attribute shift (*i.e.*, age, year, and brightness). Further, Figure 4 presents the model's prediction confidence across varying degrees of the attribute shift.

**Image.**    By gradually changing the age, NAtS samples move toward the space between the two clusters of ID samples (*i.e.*, the decision boundary) as can be seen in Figure 3 [Top]. Further, Figure 4a demonstrates that confidence decrease as we increase the degree of attribute shift, which indicates that NAtS samples move close to the decision boundary. Note that the majority of the NAtS samples still overlap with ID sample clusters as we change the age.

**Text.**    As shown in Figure 3 [Middle], NAtS samples gradually move away from the ID samples (and away from the decision boundary) as the year changes. In contrast to the UTKFace dataset, the confidence gradually increases, as shown in Figure 4b, since the NAtS samples are getting far away from the decision boundary.

| | UTKFace | | Amazon Review | | RSNA Bone Age | |
|---|---|---|---|---|---|---|
| | CE | Ours | CE | Ours | CE | Ours |
| Accuracy | $94.6\pm0.6$ | $94.1\pm0.7$ | $85.3\pm0.9$ | $84.0\pm0.9$ | $93.0\pm0.6$ | $92.3\pm0.3$ |

Table 1: In-distribution classification accuracy on three datasets with cross-entropy loss and our proposed loss.

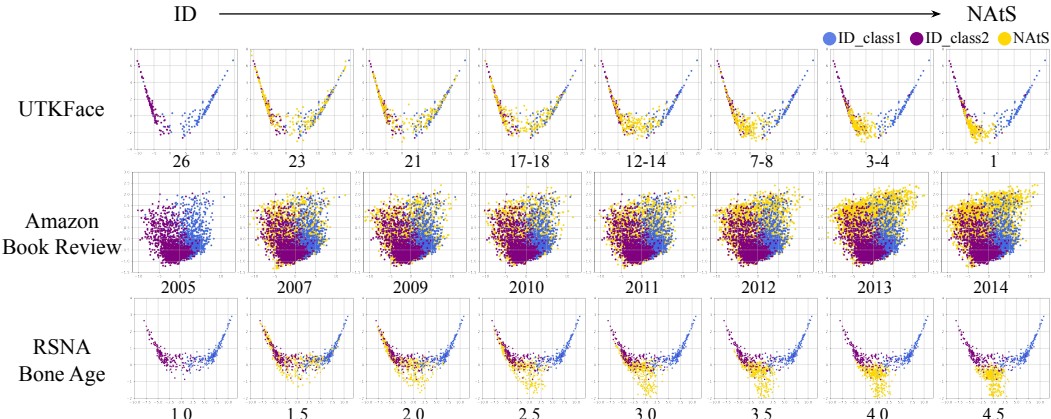

Figure 3: PCA visualization to demonstrate the movement of NAtS samples as we vary the age, year and brightness in UTKFace, Amazon Review and RSNA Bone Age dataset respectively.

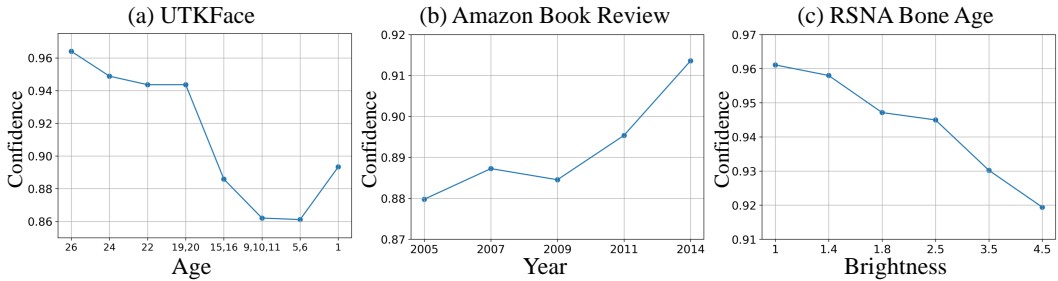

Figure 4: Impact on prediction confidence on varying age, year, and brightness in UTKFace, Amazon Review, and RSNA Bone Age dataset, respectively.

**Medical.** Figure 3 [Bottom] demonstrates that when we increase the brightness, NAtS samples move to the middle of the two classes and also move towards the outer edge of the ID sample clusters. Furthermore, as shown in Figure 4c, the relatively decreased confidence indicates that NAtS samples are placed near the decision boundary as we increase the brightness of the images.

## 5.2 COMPARISON BETWEEN CONFIDENCE-BASED AND DISTANCE-BASED OOD DETECTION

**Confidence-based methods.** Figure 2 and Figure 3 illustrate that confidence-based methods achieve high AUROC when the NAtS samples are near the decision boundary due to their low confidence. In the image and medical domain, we observed the NAtS samples moving towards the decision boundary with increasing attribute shift, thus they are detected by the confidence-based methods. In contrast, in the text domain, the high prediction confidence of the shifted NAtS samples causes the degradation in the AUROC of the confidence-based methods. Therefore, we conclude that to effectively utilize the confidence-based methods in all three NAtS datasets, it is necessary to reduce the confidence of samples outside the ID, namely, enforce NAtS samples to move near the decision boundary, which is not always possible (*e.g.,* Amazon Review dataset).

**Distance-based methods.** From Figure 2 and Figure 3, we observe that the distance-based OOD detection method (*i.e.*, Mahalanobis Detector) achieves high AUROC when NAtS samples are sufficiently away from the ID samples. In the text and medical domains, the Mahalanobis detector worked well since NAtS samples moved sufficiently away from the ID samples as the shift increased. However, in the image domain, the method fails to detect NAtS samples because instead of deviating from the ID, they move intermediately between the classes.

Prior works (Luo et al., 2020; Liu et al., 2017) report that the cross-entropy loss cannot guarantee a sufficient inter-class distance. In other words, representations do not need to be far from the decision boundary to lower the cross-entropy loss. In this regard, we assume that the performance degradation of the Mahalanobis detector is caused by the cross-entropy loss learning latent features that are not separable enough to detect the NAtS samples located between the classes (*e.g.*, Figure 3 [Top]). Specifically, if some classes are located nearby in the feature space, samples moving between classes (*i.e.*, the case of the UTKFace dataset) will not be far from the ID. Even though NAtS samples move away from one of the ID class cluster, they will gradually get closer to another ID class cluster.

## 5.3 NATS CATEGORIZATION

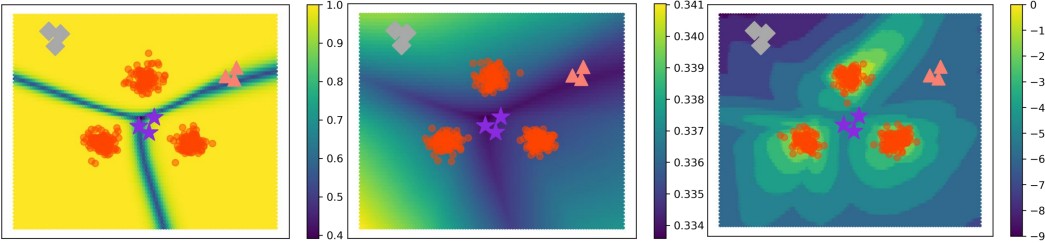

Figure 5: ID score landscape (brighter region means higher ID score) of the existing OOD detection methods (left: MSP, middle: ODIN, right: Mahalanobis). We use a synthetic 2D dataset to train a 4-layer ResNet. The **Red points** represent the ID samples; **Purple Stars**, **Gray Diamonds** and **Orange Triangles** indicate samples from different NAtS categories. A sample is regarded as NAtS when it has a low ID score.

Considering the performance of confidence-based and distance-based detection methods, we now divided NAtS into three categories based on two criteria: 1) whether the dataset is near the decision boundary or not; 2) whether they are close or far from the in-distribution dataset. Since a dataset (*i.e.*, samples) far from the decision boundary and overlapping with the in-distribution dataset is not a distributionally shifted dataset, our work focuses on the remaining three cases that cover all possible scenarios of NAtS. Without loss of generality, we use a classification task with three classes as a motivating example depicted by Figure 5.

- **NAtS category 1:** This category comprises of NAtS samples that are located near the decision boundary, and between ID samples of different classes. **Purple Stars** in Figure 5 represents samples from this category. Such samples are easily detected by confidence-based methods, MSP and ODIN, but harder to be detected by Mahalanobis Detector, which is a distance-based method.

- **NAtS category 2:** This category consists of NAtS samples that are placed away from the decision boundary and the ID data. For example, **Gray Diamonds** in Figure 5. Mahalanobis detector regards such samples as NAtS, whereas confidence-based methods fail to detect them since NAtS samples have a higher prediction confidence (*i.e.*, higher ID score) than ID samples.

- **NAtS category 3:** This category mainly comprises of NAtS samples located anywhere near the decision boundary but far away from ID data. For example, **Orange Triangles** in Figure 5. Such samples are easily detected by both distance-based and confidence-based OOD detection methods.

|  | Category 1 | Category 2 | Category 3 |
|---|---|---|---|
| Decision Boundary | Near | Far | Near |
| In-distribution | Near | Far | Far |
| Confidence-based methods works? | ✓ | ✗ | ✓ |
| Distance-based methods works? | ✗ | ✓ | ✓ |

Table 2: Comparison between different categories of NAtS based on the location of samples in the features space and performance of confidence-based and distance-based methods.

## 6 METHOD FOR CONSISTENT NATS DETECTION PERFORMANCE

In this section, we suggest a modification in the training objective for deep classifiers to encourage consistent NAtS detection performance on all NAtS categories. Then we provide experiment results where the proposed method was compared against diverse OOD detection methods on the three NAtS datasets (UTKFace, Amazon Review, RSNA Bone Age).

### 6.1 METHOD

For a generally applicable OOD detection method to all NAtS categories, we suggest a new training objective for deep classifiers comprised of *classification loss* ($\mathcal{L}_{\text{CE}}$), *distance loss* ($\mathcal{L}_{\text{dist}}$) and *entropy loss* ($\mathcal{L}_{\text{entropy}}$). The proposed objective improves the performance of the Mahalanobis detector on NAtS samples from category 1 without sacrificing performance on NAtS samples from other categories. We focus on improving the distance-based OOD detection method rather than the confidence-based method since it is not always possible to enforce NAtS samples to be embedded near the decision boundary, as discussed in Section 5.2.

The proposed training loss is defined as:

$$\mathcal{L}_{\text{total}} = \mathcal{L}_{\text{CE}} + \mathcal{L}_{\text{dist}} + \mathcal{L}_{\text{entropy}}. \tag{1}$$

Note that for classification loss, we use the standard cross-entropy loss, but other losses such as focal loss can also be used. The distance loss is used to increase the distance between distinct class distributions of ID samples so that NAtS samples have larger space to move around without overlapping with ID clusters, especially in NAtS category 1:

$$\mathcal{L}_{\text{dist}} = \lambda_1 \frac{1}{\binom{K}{2}} \sum_l \sum_{k \neq l} \frac{\|\mu_l - \mu_k\|_2}{\sqrt{D}}, \tag{2}$$

where $\lambda_1 < 0$ is a hyperparameter, $K$ is the number of target classes, $D$ is the dimension of feature representation in the latent space (typically the penultimate layer), and $\mu_l$ is the mean vector of the features of samples with label $l$. Since the value of the vector norm increases as the dimension of the feature space increases, we normalize the distance by the square root of the feature dimension.

As will be discussed in Section D of Appendix, we discovered after some initial experiments that the *distance loss* often made the model use a very limited number of latent dimensions to increase the distance between class mean vectors, which degraded the NAtS detection performance. In other words, adding only $\mathcal{L}_{\text{dist}}$ to $\mathcal{L}_{\text{CE}}$ caused the latent feature space to collapse into a very small number of dimensions (*i.e.* rank-deficient), which caused all NAtS samples to be embedded near the ID samples. Therefore, we add *entropy loss* to increase the number of features used to represent samples.

The *entropy loss* is defined as:

$$\mathcal{L}_{\text{entropy}} = \lambda_2 \frac{D}{\sum_i \text{Var}(\mathbf{z}_i)} + \lambda_3 \frac{1}{\binom{D}{2}} \sum_i \sum_{j \neq i} C_{ij}^2, \tag{3}$$

where $\lambda_2 > 0$ and $\lambda_3 > 0$ are hyperparameters, $\mathbf{z} \in \mathbb{R}^D$ is the feature representation in the latent feature space, $\text{Var}(\cdot)$ is variance, and $C_{ij}$ is the correlation coefficient between $i$th and $j$th dimension of the feature space. Specifically, $C_{ij}$ is described as:

$$C_{ij} = \frac{\text{Cov}(\mathbf{z}_i, \mathbf{z}_j)}{\sigma(\mathbf{z}_i)\sigma(\mathbf{z}_j)}, \tag{4}$$

where $\text{Cov}(\cdot)$ and $\sigma(\cdot)$ are covariance and the standard deviation, respectively.

Intuitively, the first term in equation 3 encourages each latent dimension to have diverse values, therefore preventing the latent feature space from collapsing into a confined space. With the first term alone, however, all latent dimensions might learn correlated information, thus making the latent space rank-deficient. Therefore, we use the second term in equation 3 to minimize the correlation between different latent dimensions. Note that minimizing the feature correlation was also used in previous works under different contexts such as self-supervised learning (Zbontar et al., 2021).

## 6.2 RESULTS AND DISCUSSION

To demonstrate the effectiveness of the suggested method, we train all classifiers using the standard cross-entropy loss and our modified loss and compare post-hoc OOD detection methods across three NAtS datasets. Specifically, we present the results of the confidence-based (*i.e.*, MSP and ODIN) and the distance-based methods (*i.e.*, Mahalanobis distance). We also include a recently proposed OOD detection method (Chandramouli & Sageev, 2020) which computes channel-wise correlations in CNN with the gram matrix and estimates the deviation of the test samples from the training samples to detect the OOD samples. [1] Although this method uses the distance in the channel correlation space, we expect it to behave more similarly to the Mahalanobis detector than confidence-based methods, namely MSP and ODIN. We also compare with another recent baseline which exploits the energy score to detect OOD samples (Liu et al., 2020). As the method leverage the logit layer to calculate the energy score, and the softmax score is based on the logit values, we conjecture that the energy score demonstrates detection ability similar to the confidence-based methods.

---

[1]However, in the text domain, the notion of gram matrix is vague, and hence we do not compare against this baseline.

| Distribution | Age | CE | | | | | Ours |
| --- | --- | --- | --- | --- | --- | --- | --- |
| | | MSP | ODIN | Mahalanobis | Gram matrix | Energy score | Mahalanobis |
| ID | 26 | – | – | – | – | – | – |
| NAtS | 25 | 49.7±1.8 | 46.8±0.9 | 50.0±1.2 | 51.5±1.0 | 49.5±1.5 | 50.1±1.7 |
| | 24 | 52.6±1.6 | 51.3±0.7 | 50.6±2.8 | 52.0±1.1 | 52.4±1.5 | 53.3±1.3 |
| | 23 | 50.1±1.4 | 48.5±0.8 | 47.9±2.4 | 52.3±1.3 | 49.8±1.2 | 52.7±1.2 |
| | 22 | 52.5±1.6 | 49.7±1.0 | 51.2±1.7 | 53.2±1.7 | 52.3±1.1 | 56.5±1.3 |
| | 21 | 53.8±1.7 | 51.3±1.0 | 49.5±2.4 | 52.5±1.5 | 53.4±1.7 | 55.3±1.0 |
| | 19-20 | 55.9±2.1 | 51.8±1.0 | 52.2±1.5 | 52.4±1.6 | 55.7±1.8 | 57.9±1.2 |
| | 17-18 | 62.4±1.6 | 56.7±1.0 | 52.9±1.6 | 54.0±0.7 | 62.0±1.3 | 61.0±1.1 |
| | 15-16 | 69.4±1.5 | 60.1±2.0 | 59.9±2.9 | 56.6±1.6 | 68.7±1.1 | 70.7±0.7 |
| | 12-14 | 76.1±1.3 | 69.2±2.3 | 57.7±2.6 | 57.7±1.6 | 75.5±1.1 | 75.3±0.9 |
| | 9-11 | 77.9±0.7 | 77.4±2.6 | 56.0±3.8 | 55.9±2.6 | 77.4±1.5 | 80.5±0.9 |
| | 7-8 | 77.6±0.5 | 78.7±2.1 | 53.3±4.4 | 56.1±1.4 | 77.0±1.4 | 82.1±1.9 |
| | 5-6 | 79.6±1.0 | 80.5±2.6 | 53.2±2.4 | 55.4±2.4 | 79.2±1.2 | 83.5±1.4 |
| | 3-4 | 79.8±1.2 | 85.5±2.4 | 56.4±3.2 | 55.4±1.2 | 79.0±1.5 | 88.6±1.0 |
| | 2 | 79.3±0.6 | 88.1±1.7 | 53.3±2.8 | 53.5±0.8 | 78.6±1.8 | 88.3±1.6 |
| | 1 | 80.7±0.4 | 90.7±0.9 | 51.2±4.4 | 52.9±2.1 | 79.9±1.3 | 90.3±2.0 |

*(Distribution: UTKFace)*

| Distribution | Year | CE | | | | | Ours |
| --- | --- | --- | --- | --- | --- | --- | --- |
| | | MSP | ODIN | Mahalanobis | Gram matrix | Energy score | Mahalanobis |
| ID | 2005 | – | – | – | – | – | – |
| NAtS | 2006 | 49.6±0.4 | 49.6±0.4 | 51.5±0.1 | - | 49.5±0.2 | 51.4±0.1 |
| | 2007 | 49.0±1.4 | 49.0±1.4 | 56.8±0.2 | - | 47.7±0.7 | 56.5±0.1 |
| | 2008 | 48.4±1.1 | 48.4±1.1 | 55.2±0.3 | - | 47.5±0.5 | 54.9±0.1 |
| | 2009 | 48.5±1.1 | 48.5±1.1 | 53.7±0.2 | - | 47.8±0.5 | 53.6±0.1 |
| | 2010 | 48.1±0.2 | 48.7±1.2 | 54.0±0.5 | - | 47.9±0.6 | 53.6±0.1 |
| | 2011 | 48.1±0.3 | 47.3±1.6 | 55.6±0.7 | - | 45.7±0.4 | 54.9±0.0 |
| | 2012 | 45.7±0.3 | 45.5±2.4 | 63.3±0.8 | - | 42.0±0.9 | 62.6±0.1 |
| | 2013 | 38.2±0.6 | 43.0±3.5 | 75.5±0.5 | - | 38.2±0.9 | 75.1±0.0 |
| | 2014 | 38.9±0.8 | 43.7±3.7 | 76.8±0.2 | - | 38.8±1.0 | 76.7±0.1 |

*(Distribution: Amazon Review)*

| Distribution | Brightness | CE | | | | | Ours |
| --- | --- | --- | --- | --- | --- | --- | --- |
| | | MSP | ODIN | Mahalanobis | Gram matrix | Energy score | Mahalanobis |
| NAtS | 0.0 | 93.9±4.2 | 100.0±0.0 | 99.9±0.2 | 99.8±0.5 | 92.2±4.1 | 100.0±0.0 |
| | 0.2 | 71.7±3.5 | 89.1±6.1 | 93.7±1.8 | 68.7±2.7 | 71.0±4.3 | 98.0±1.0 |
| | 0.4 | 55.7±2.2 | 64.5±4.8 | 79.5±4.5 | 54.6±3.3 | 55.2±2.6 | 89.8±2.2 |
| | 0.6 | 52.3±1.5 | 53.4±3.1 | 64.5±5.5 | 51.3±1.9 | 52.1±1.6 | 74.6±2.5 |
| | 0.8 | 50.2±1.1 | 48.9±1.8 | 53.3±2.4 | 49.9±1.0 | 50.1±1.0 | 56.3±1.8 |
| ID | 1.0 | – | – | – | – | – | – |
| NAtS | 1.2 | 51.6±1.2 | 54.9±1.7 | 58.5±1.9 | 51.4±1.0 | 51.4±1.3 | 55.0±2.4 |
| | 1.4 | 54.7±2.2 | 63.5±3.0 | 69.0±3.1 | 54.8±1.5 | 54.4±2.3 | 65.1±3.1 |
| | 1.6 | 58.5±3.5 | 71.7±4.0 | 76.9±4.0 | 59.8±1.9 | 57.9±3.3 | 75.5±3.9 |
| | 1.8 | 61.1±3.9 | 80.1±4.5 | 82.1±5.1 | 62.6±2.0 | 60.5±3.7 | 82.3±4.3 |
| | 2.0 | 62.5±4.3 | 87.1±4.1 | 85.8±5.5 | 64.2±3.5 | 61.8±4.2 | 87.0±4.0 |
| | 2.5 | 66.5±5.3 | 95.0±2.3 | 91.9±5.0 | 69.5±4.6 | 65.8±5.3 | 93.7±2.9 |
| | 3.0 | 75.1±3.9 | 98.2±0.9 | 95.4±3.3 | 76.4±4.0 | 74.6±3.6 | 96.6±1.8 |
| | 3.5 | 80.1±2.5 | 99.2±0.4 | 96.8±2.7 | 80.3±5.1 | 79.6±1.9 | 97.8±1.3 |
| | 4.0 | 83.4±2.5 | 99.6±0.3 | 97.7±2.1 | 84.7±5.7 | 82.9±1.8 | 98.4±0.9 |
| | 4.5 | 85.1±3.5 | 99.7±0.3 | 98.2±1.6 | 87.8±4.9 | 84.5±2.8 | 98.7±0.8 |

*(Distribution: RSNA Bone Age)*

Table 3: NAtS detection performance on distributional shifts in three datasets measured by AUROC.

For a fair comparison, we use only the penultimate layer for evaluating the Mahalanobis detector and Gram matrix since our method is developed based on the analysis of the NAtS samples in the penultimate layer feature space. We also provide results of the ensemble version that utilize all layers in Section E of the Appendix. We describe the details of experiment setups and selecting the values for $\lambda_1$, $\lambda_2$ and $\lambda_3$ in the Appendix Section C, where the three values can be reasonably chosen without any explicit NAtS validation datasets. We also present an ablation study to investigate the effect of different terms in the proposed loss in Section D of the Appendix. We also compare with other recent baselines that suggest other training objective for OOD detection in Section E of the Appendix.

As shown in Table 1, the in-distribution classification accuracy of the model trained with our suggested loss is comparable to that of the model trained with the cross-entropy loss. Further, we present the NAtS detection performance of the baselines and our method in Table 3. As we expected above, the gram matrix achieves performance similar to the Mahalanobis detector, demonstrating the low AUROC in the UTKFace dataset. Interestingly, even though the energy score is calculated based on

the logit values, which are more tempered than the softmax score, it shows NAtS detection performance similar to that of MSP, which is a confidence-based method. These results show that similar to MSP, ODIN, and Mahalanobis, Gram matrix and Energy-based methods also have performance inconsistency on NAtS datasets. We report the additional NAtS detection performance on four other metrics, which are often used in the OOD detection community in Section F of Appendix.

Also, it is readily visible that our proposed training objective makes the Mahalanobis detector a robust NAtS detection method for all NAtS categories. In UTKFace, we can see the dramatic NAtS detection performance increase for Mahalanobis detector. And for the other two datasets, the proposed loss does not decrease the NAtS detection performance of Mahalanobis detector. Note that ODIN is more sensitive to OOD samples than Mahalanobis detector for some brightness levels in the RSNA Bone Age dataset, but it shows inconsistent performance across all three datasets.

## 7    RELATED WORK

**Extensive evaluation on distribution shifts.**    Recent works have leveraged the distribution shifts for OOD detection and uncertainty estimation. Rabanser et al. (2019) primarily focus on detecting the entire shifted distribution, not on detecting a single shifted sample. Yaniv et al. (2019) quantify the predictive uncertainty and investigate the quality of the calibration under the dataset shift. Engkarat & Takayuki (2019) aims to predict the classification accuracy of a shifted distribution by utilizing the average OOD score of the distribution. Hsu et al. (2020) also consider the distribution shift with preserved label space. However, Hsu et al. (2020) focus on the shifted distribution from different domain (*e.g.*, real images vs. sketches ). To the best of our knowledge, none of these works conduct an *in-depth analysis* on the performance of existing OOD detection methods on samples shifted based on natural attributes.

**OOD detection methods.**    Post-hoc OOD detection methods (Hendrycks & Gimpel, 2017; Liang et al., 2018; Lee et al., 2018; Chandramouli & Sageev, 2020) that utilize the classification models to obtain the OOD scores have achieved remarkable performance. All of these works aim to detect shifted sample which might affect decision making system in terms of confidence and hidden representation. Other OOD detection approach is to train the generative model (Ren et al., 2019; Choi et al., 2018; Mahmood et al., 2020) on the training distribution and estimate the density of OOD samples in test time. While these approaches are viable, it is not directly related with the downstream task but aims to detect features which is different from training distribution. In this paper, we mainly focus on methods that utilize the classification models to detect OOD samples since we are mainly interested in samples that affect decision-making systems.

**Model uncertainty.**    A number of previous works measure model uncertainty using various methods such as Bayesian neural networks (Blundell et al., 2015), Monte Carlo dropout (Gal & Ghahramani, 2016), and deep ensembles (Lakshminarayanan et al., 2017). Note that technically, model uncertainty can be used to detect NAtS samples, especially for NAtS categories 1 and 3, since sampling model weights from the function space can be seen as redrawing the decision boundary, and NAtS samples in categories 1 and 3 will be affected heavily by this process. Model uncertainty, however, aims to capture the uncertainty in the model weights rather than detecting OOD samples, making the two rather independent research directions.

## 8    CONCLUSIONS

To enhance the reliability of decision-making systems, we define a new task, Natural Attribute-based Shift (NAtS) detection, that aims to detect the samples shifted by a natural attribute. We introduce NAtS detection benchmark datasets by adjusting the natural attributes present in the existing datasets. Through extensive evaluation of existing OOD detection methods on NAtS datasets, we observe inconsistent performance depending on the nature of NAtS samples. Then, we analyze the inconsistency by probing the relationship between the location of NAtS samples and the performance of existing OOD detection methods. Based on this observation, we suggest a simple remedy to help Mahalanobis OOD detection method to have consistent performance across all NAtS categories. We hope our dataset and task inspire fellow researchers to investigate practical methods for identifying NAtS, which is crucial for deploying the prediction models in real-world systems.

ETHICS STATEMENT

We believe this is not applicable to us.

REPRODUCIBILITY STATEMENT

We describe the implementation and training details along with the details on computation resources in Section C of the Appendix. We submit our source codes along with the instructions to reproduce our results in the Appendix.

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

# Table of Contents

## A  DETAILS OF OOD DETECTION METHODS

In this section, we describe the three post-hoc and task-agnostic OOD detection methods in detail, focusing mainly on their formulation and how each method assigns an OOD score to an input sample.

### A.1  MAXIMUM OF SOFTMAX PROBABILITY (MSP)

In this method, the maximum of softmax probability is considered as confidence score (Hendrycks & Gimpel, 2017). Formally, we calculate the maximum softmax probability as follows:

$$S_{\text{MSP}}(\mathbf{x}) = \max_c \frac{\exp\left(\mathbf{z}^{(c)}\right)}{\sum_{j=1}^{C} \exp\left(\mathbf{z}^{(j)}\right)},$$

where $C$ is the number of target classes, $c$ is the index of a class, and $\mathbf{z}^{(j)}$ denotes $j^{th}$ attribute of the feature in the logit layer.

### A.2  ODIN

ODIN (Liang et al., 2018) utilized two well established techniques, namely temperature scaling and input preprocessing to increase the difference between softmax scores of in-distribution and OOD samples. Temperature scaling was originally proposed in Hinton et al. (2015) to distill the knowledge in neural networks and was later adopted widely in classification tasks to calibrate confidence of prediction(Guo et al., 2017). In addition to temperature scaling, the input is preprocessed in order to increase the softmax score of given input by adding small perturbations which are obtained by back-propagating the gradient of the loss with respect to the input. More specifically, ODIN is computed as follows:

$$S(\mathbf{x}; T) = \max_c \frac{\exp\left(\mathbf{z}^{(c)}/T\right)}{\sum_{j=1}^{C} \exp\left(\mathbf{z}^{(j)}/T\right)},$$

where $T \in \mathbb{R}^+$ is the temperature scaling parameter, $C$ is the number of target classes, $c$ is the index of class, and $\mathbf{z}^{(j)}$ denotes $j^{th}$ attribute of the logit layer features of input $\mathbf{x}$. During training, $T$ is set to 1.

For OOD detection, the input is first pre-processed as follows:

$$\tilde{\mathbf{x}} = \mathbf{x} - \varepsilon \operatorname{sign}\left(-\nabla_{\boldsymbol{x}} \log S(\mathbf{x}; T)\right),$$

where $\varepsilon$ represents the magnitude of perturbation.

Next, the network calculates the calibrated softmax score of the preprocessed input as follows:

$$S_{\text{ODIN}}(\mathbf{x}; T) = \max_c \frac{\exp\left(\tilde{\mathbf{z}}^{(c)}/T\right)}{\sum_{j=1}^{C} \exp\left(\tilde{\mathbf{z}}^{(j)}/T\right)},$$

where $\tilde{\mathbf{z}}^{(j)}$ denotes $j^{th}$ attribute of the logit layer features of the preprocessed input $\tilde{\mathbf{x}}$ .

Lastly, the modified softmax score is compared to a threshold value $\delta$. If the score is greater than the threshold, then the input is classified as ID sample and otherwise OOD. Originally, $T$, $\varepsilon$, and $\delta$ are hyperparameters and are selected such that the false positive rate (FPR) at true positive rate (TPR) 95% is minimized on the validation OOD dataset. However, the performance saturates when $T$ is greater than 1000 and therefore, in general, a large value of $T$ is preferred. Following this, in this paper, we fix $T = 1000$ for our experiments.

### A.3 MAHALANOBIS DETECTOR

To obtain Mahalanobis distance-based OOD score (Lee et al., 2018) of a sample, we calculate the mahalanobis distance from the clusters of classes to the sample. Then, the distance from the closest class is chosen as the confidence score.

Specifically, the Mahalanobis score of an input $\mathbf{x}$ is defined as

$$S_{\text{mahala}}(\mathbf{x}) = \sum_l \alpha_l \max_c - \left(f_\theta^l(\mathbf{x}) - \mu_{c,l}\right)^\top \mathbf{\Sigma}_l^{-1}\left(f_\theta^l(\mathbf{x}) - \mu_{c,l}\right),$$

where $c$ and $l$ are the class and layer index, respectively, $f_\theta^l$ is the $l^{th}$ layer's feature representation of an input $\mathbf{x}$, $\mu_{c,l}$ and $\mathbf{\Sigma}_l$ are their class mean vector and tied covariance of the training data, correspondingly.

Note that **ODIN** and **Mahalanobis Detector** assume the availability of OOD validation dataset. However, some recent works (Shafaei et al., 2019; Techapanurak et al., 2020) report that this assumption limits the OOD detection generalizability since a model is biased towards an OOD validation set. In response, this paper validate the performance of OOD methods in the version that does not require to tune with OOD validation dataset. We perform the performance of ODIN as Hsu et al. (2020). We do not perform Mahalanobis Detector ensembling over on all layers with the optimal linear combination which requires explicit OOD data. Instead, we perform two version that use only the penultimate layer of hidden representation and sum uniformly over all layers. Therefore, for all our experiments, we use modified OOD detection methods that do not require the OOD validation dataset.

## B   ANALYSIS ON THE TEXT DATASET

In this section, we provide a detailed analysis of the text dataset. We use the Amazon Review (He & McAuley, 2016; McAuley et al., 2015) dataset. We consider the product category "book" and conducted an analysis to see the impact of time on product reviews. We then performed an analysis to see the impact of time on the length of the reviews. Figure 6 presents a comparison between density plot of review length for each year from 2006 to 2014. We observe that as we move ahead in time,

the length of the reviews gradually reduces. Further, Figure 7 presents the distribution of the average ratio of important words in a sequence by year. First, we train a model to classify sentiment polarity using Catboost (Prokhorenkova et al., 2017) using document term frequency vector. We then extract top-100 important words using feature importance. Lastly, we manually select important words over 100 words. We find that the distribution of ratio of important words in a sequence gradually increase over time. Based on this analysis, we figure out that the feature related with downtream task is shifted by time.

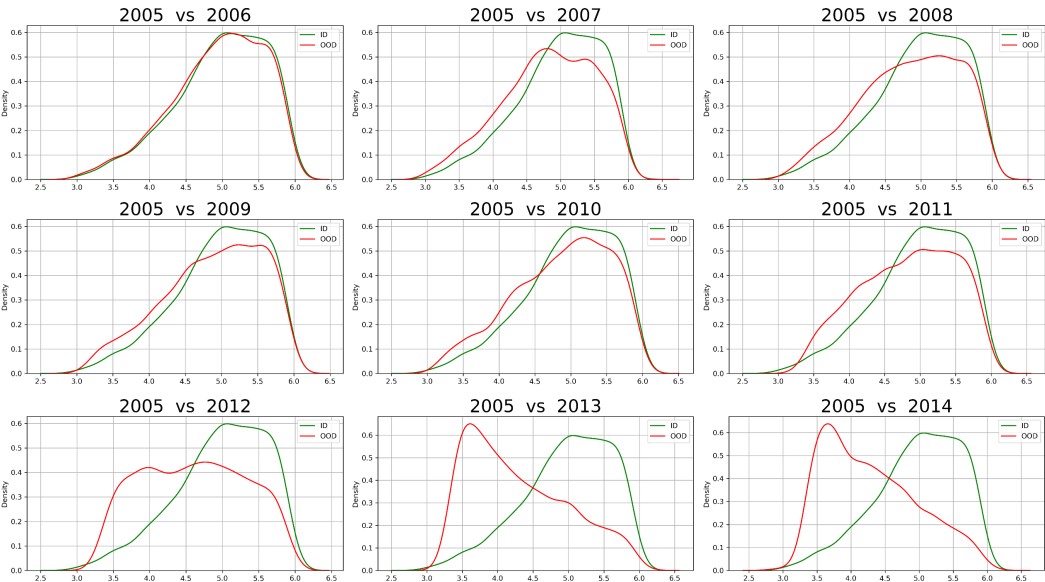

Figure 6: The density plot of sequence length per year. The x-axis represents the length with log scaling.

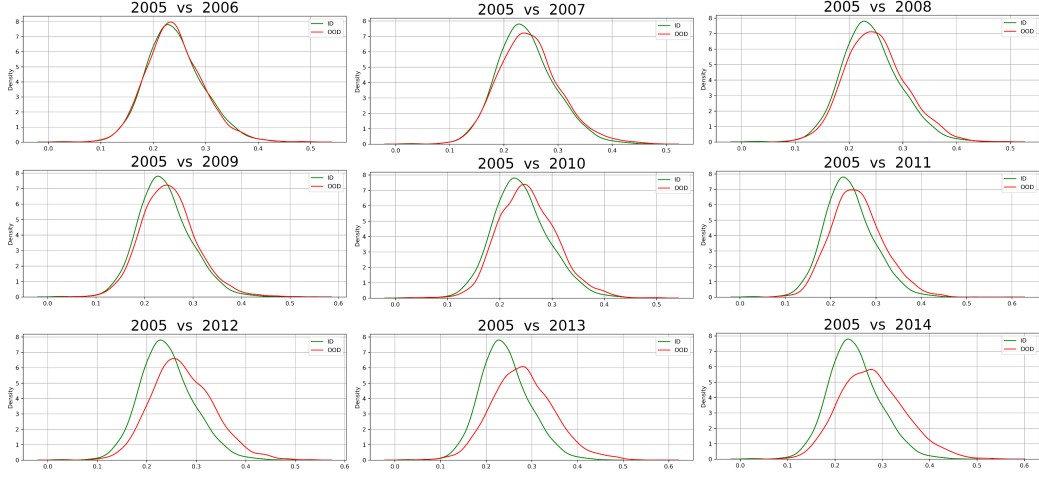

Figure 7: The density plot of the ratio of important words per sequence.

## C IMPLEMENTATION DETAILS

In this section, we describe the training details. Followed by it, we describe the algorithm we used to select the hyperparameters in our suggested modification to the loss function. Then, we provide the links to download the datasets.

## C.1 TRAINING DETAILS AND COMPUTING INFRASTRUCTURE

**Image.** We used ResNet18 (He et al., 2016) that is pretrained with ImageNet (Deng et al., 2009) to train a gender classifier using a UTKFace NAtS dataset. After the average pooling layer, we trained a fully-connected network composed of 2 hidden layers which have 128 and 2 units respectively with a relu activation. The network is trained for 100 epochs with a batch size of 64. We used stochastic gradient descent with Adam optimizer (Kingma & Ba, 2014) and set learning rate as $3e - 5$. For data augmentation, the technique of SimCLR (Chen et al., 2020) is used for all the experiments.

**Text.** We perform experiment with Transformer network with 4 layers. The network is trained for 10 epochs with batch size of 128. We used Adam as an optimizer and set the learning rate as $3e - 6$.

**Medical.** We use a ResNet18 (He et al., 2016) model, pretrained on ImageNet (Deng et al., 2009), and add two fully-connected layers containing 128 and 2 hidden units with a relu activation. We train the network to classify gender given the x-ray image. Each model is trained for 30 epochs using SGD optimizer with a learning rate of 0.01 and momentum of 0.9, using a batch size of 64.

All the experiments are conducted on a single GeForce RTX 3090 GPU with 24GB memory. The results are measured by computing mean and standard deviation across 5 trials upon randomly chosen seeds.

## C.2 HYPERPARAMETER TUNING

Our suggested modification to the training loss mainly comprises of *distance loss* ($\mathcal{L}_{\text{dist}}$) and a *entropy loss* ($\mathcal{L}_{\text{entropy}}$). More formally, the training loss is given by:

$$\mathcal{L}_{\text{total}} = \mathcal{L}_{\text{CE}} + \mathcal{L}_{\text{dist}} + \mathcal{L}_{\text{entropy}}. \tag{5}$$

The *entropy loss* comprises of two terms and is defined as:

$$\mathcal{L}_{\text{entropy}} = \lambda_2 \frac{D}{\sum_i \text{Var}(\mathbf{z}_i)} + \lambda_3 \frac{1}{\binom{D}{2}} \sum_i \sum_{j \neq i} C_{ij}^2, \tag{6}$$

where $\lambda_2 > 0$ and $\lambda_3 > 0$ are hyperparameters, $\mathbf{z} \in \mathbb{R}^D$ is the feature representation in the latent feature space, and $C_{ij}$ is the correlation coefficient between $i$th and $j$th dimension of the feature space. For more details, please refer to Section 6.1.

For simplicity, in this section, we will refer the first term of *entropy loss* as *variance loss*, and the second term as *correlation loss*. To obtain the hyperparameters of different terms in our loss function, we explore the value of $\lambda_2$ in $[0.01, 0.1, 1.0, 10.]$ and $\lambda_3$ in $[0.0001, 0.001, 0.01, 0.1, 1.0]$. The hyperparameter corresponding the *distance loss*, $\lambda_1$, is set as 0.1, and then the hyperparameters of the *variance loss* and the *correlation loss* are chosen by a simple algorithm.

**Algorithm:** We now describe the algorithm we used to find the hyperparameters of *variance* and *correlation loss*. First, we calculate the harmonic mean of the *variance loss* and *correlation loss* using our training dataset. Next, we select the hyperparameters with the lowest value of harmonic mean. Then, to ensure that *entropy loss* prevents the feature space collapsing problem, we apply the singular value decomposition (SVD) in the penultimate features and test if the sum of singular values except the two largest values is improved by the *entropy loss* or not. Concretely, to prove the enhancement, we compare the values with those of the model trained with the $\mathcal{L}_{\text{CE}} + \lambda_1 \mathcal{L}_{\text{dist}}$. If the selected hyperparameters do not improve the values, we reject them and investigate the hyperparameters that have the next lowest harmonic mean. The hyperparameters satisfying the above steps are selected as the hyperparameters of our loss function.

Note that we reject the hyperparameters if they significantly degrade the classification accuracy. In image domain, we applied the algorithm and obtained 0.1, 0.1, 0.0001 as hyperparameters of $\lambda_1$, $\lambda_2$, and $\lambda_3$, respectively. In the medical domain, we obtain the hyperparameter corresponding to $\lambda_3$ as 1.0, and the other hyperparameters are the same as those of the image domain. In the text domain, 10. and 1.0 are used for the $\lambda_2$ and $\lambda_3$. The *distance loss* is set by 0.1.

## C.3 Links for Datasets

In our work, we use the openly available datasets, namely, UTKFace dataset[2], Amazon Review dataset[3], and RSNA Bone Age dataset[4].

# D   Ablation Studies on Distance and Entropy Loss

In this section, we conduct an ablation study to investigate the effect of each proposed term in the loss function in  equation 5. Table 4 presents an ablation study to find the effect of the *entropy loss* and *distance loss* in the proposed modification to training loss. In UTKFace, when training with solely attaching $\mathcal{L}_{\text{dist}}$ or $\mathcal{L}_{\text{entropy}}$ to $\mathcal{L}_{\text{CE}}$, the NAtS detection performance using mahalanobis OOD detection method improves compared to training with only $\mathcal{L}_{\text{CE}}$, but the performance does not monotonically increase in both cases along with the variation of Age. However, when both $\mathcal{L}_{\text{dist}}$ and $\mathcal{L}_{\text{entropy}}$ are used, there is a large improvement in performance with the results being monotonically increasing as we move from ID towards NAtS, which implies that these two losses are mutually beneficial. In the text and medical domains, although there is a little degradation of AUROC when only $\mathcal{L}_{\text{dist}}$ is added, the performance is improved by using $\mathcal{L}_{\text{entropy}}$ in the training objective.

Further, to analyze the impact of each loss term to the feature-level representations, we perform a singular value decomposition (SVD) in the penultimate layer of ResNet18 trained on the UTKFace dataset, similar to Verma et al. (2019). When trained with $\mathcal{L}_{\text{CE}}$ only, the sum of the two largest singular values was 617.47, while the sum of the remaining singular values was 785.54. When trained with $\mathcal{L}_{\text{CE}} + \mathcal{L}_{\text{dist}}$, the sum of the two largest singular values increased (4481.88), while the sum of the remaining singular values decreased (371.86).

As discussed in Section 6.1 in the main paper, this indicates that the addition of the *distance loss* is likely to collapse the latent feature space (*i.e.* rank-deficient), thus possibly decreasing the model's sensitivity to NAtS samples. For example, if a model is trained to classify apples and banaNAtS based only on the color, it will not be able to tell that a firetruck is a NAtS sample. When we train the classifier with $\mathcal{L}_{\text{entropy}}$ added, the problem is effectively alleviated. The sum of the two largest singular value is reduced from 4481.88 to 2960.63, and the sum of the remaining singular values increased from 371.86 to 513.13.

---

[2]https://susanqq.github.io/UTKFace/
[3]https://jmcauley.ucsd.edu/data/amazon/
[4]https://www.rsna.org/education/ai-resources-and-training/ai-image-challenge/rsna-pediatric-bone-age-challenge-2017

| | Distribution | Age | $\mathcal{L}_{\text{CE}}$ | $\mathcal{L}_{\text{CE}} + \mathcal{L}_{\text{dist}}$ | $\mathcal{L}_{\text{CE}} + \mathcal{L}_{\text{entropy}}$ | $\mathcal{L}_{\text{CE}} + \mathcal{L}_{\text{dist}} + \mathcal{L}_{\text{entropy}}$ |
|---|---|---|---|---|---|---|
| | ID | 26 | – | – | – | – |
| | | 25 | $50.0_{\pm1.2}$ | $50.3_{\pm1.0}$ | $56.8_{\pm2.4}$ | $50.1_{\pm1.7}$ |
| | | 24 | $50.6_{\pm2.8}$ | $50.8_{\pm0.8}$ | $54.6_{\pm1.7}$ | $53.3_{\pm1.3}$ |
| | | 23 | $47.9_{\pm2.4}$ | $48.8_{\pm1.2}$ | $58.1_{\pm3.3}$ | $52.7_{\pm1.2}$ |
| | | 22 | $51.2_{\pm1.7}$ | $53.4_{\pm1.0}$ | $61.5_{\pm2.8}$ | $56.5_{\pm1.3}$ |
| | | 21 | $49.5_{\pm2.4}$ | $51.7_{\pm2.3}$ | $56.0_{\pm3.6}$ | $55.3_{\pm1.0}$ |
| | | 19-20 | $52.2_{\pm1.5}$ | $54.8_{\pm2.0}$ | $56.6_{\pm3.1}$ | $57.9_{\pm1.2}$ |
| UTKFace | | 17-18 | $52.9_{\pm1.6}$ | $58.5_{\pm2.4}$ | $62.0_{\pm4.5}$ | $61.0_{\pm1.1}$ |
| | NAtS | 15-16 | $59.9_{\pm2.9}$ | $70.0_{\pm3.4}$ | $66.6_{\pm4.2}$ | $70.7_{\pm0.7}$ |
| | | 12-14 | $57.7_{\pm2.6}$ | $67.4_{\pm5.5}$ | $65.2_{\pm5.6}$ | $75.3_{\pm0.9}$ |
| | | 9-11 | $56.0_{\pm3.8}$ | $66.5_{\pm7.6}$ | $67.7_{\pm5.8}$ | $80.5_{\pm0.9}$ |
| | | 7-8 | $53.3_{\pm4.4}$ | $64.1_{\pm7.9}$ | $69.3_{\pm7.3}$ | $82.1_{\pm1.9}$ |
| | | 5-6 | $53.2_{\pm2.4}$ | $63.0_{\pm8.0}$ | $64.3_{\pm5.1}$ | $83.5_{\pm1.4}$ |
| | | 3-4 | $56.4_{\pm3.2}$ | $64.3_{\pm10.1}$ | $65.8_{\pm5.1}$ | $88.6_{\pm1.0}$ |
| | | 2 | $53.3_{\pm2.8}$ | $58.6_{\pm10.1}$ | $57.7_{\pm4.2}$ | $88.3_{\pm1.6}$ |
| | | 1 | $51.2_{\pm4.4}$ | $58.1_{\pm10.2}$ | $55.6_{\pm5.7}$ | $90.3_{\pm2.0}$ |

| | Distribution | Year | $\mathcal{L}_{\text{CE}}$ | $\mathcal{L}_{\text{CE}} + \mathcal{L}_{\text{dist}}$ | $\mathcal{L}_{\text{CE}} + \mathcal{L}_{\text{entropy}}$ | $\mathcal{L}_{\text{CE}} + \mathcal{L}_{\text{dist}} + \mathcal{L}_{\text{entropy}}$ |
|---|---|---|---|---|---|---|
| | ID | 2005 | – | – | – | – |
| | | 2006 | $51.5_{\pm0.1}$ | $51.5_{\pm0.1}$ | $51.4_{\pm0.1}$ | $51.4_{\pm0.1}$ |
| | | 2007 | $56.8_{\pm0.2}$ | $56.3_{\pm0.1}$ | $56.6_{\pm0.1}$ | $56.5_{\pm0.1}$ |
| | | 2008 | $55.2_{\pm0.3}$ | $54.8_{\pm0.1}$ | $54.9_{\pm0.1}$ | $54.9_{\pm0.1}$ |
| Amazon Review | | 2009 | $53.7_{\pm0.2}$ | $53.6_{\pm0.1}$ | $53.6_{\pm0.1}$ | $53.6_{\pm0.1}$ |
| | NAtS | 2010 | $54.0_{\pm0.5}$ | $53.6_{\pm0.1}$ | $53.6_{\pm0.1}$ | $53.6_{\pm0.1}$ |
| | | 2011 | $55.6_{\pm0.7}$ | $54.8_{\pm0.1}$ | $54.9_{\pm0.0}$ | $54.9_{\pm0.0}$ |
| | | 2012 | $63.3_{\pm0.8}$ | $62.5_{\pm0.1}$ | $62.6_{\pm0.1}$ | $62.6_{\pm0.1}$ |
| | | 2013 | $75.5_{\pm0.5}$ | $74.7_{\pm0.1}$ | $75.1_{\pm0.1}$ | $75.1_{\pm0.0}$ |
| | | 2014 | $76.8_{\pm0.2}$ | $76.3_{\pm0.1}$ | $76.8_{\pm0.1}$ | $76.7_{\pm0.1}$ |

| | Distribution | Brightness | $\mathcal{L}_{\text{CE}}$ | $\mathcal{L}_{\text{CE}} + \mathcal{L}_{\text{dist}}$ | $\mathcal{L}_{\text{CE}} + \mathcal{L}_{\text{entropy}}$ | $\mathcal{L}_{\text{CE}} + \mathcal{L}_{\text{dist}} + \mathcal{L}_{\text{entropy}}$ |
|---|---|---|---|---|---|---|
| | | 0.0 | $99.9_{\pm0.2}$ | $99.9_{\pm0.2}$ | $100.0_{\pm0.0}$ | $100.0_{\pm0.0}$ |
| | | 0.2 | $93.7_{\pm1.8}$ | $94.0_{\pm2.3}$ | $96.9_{\pm1.2}$ | $98.0_{\pm1.0}$ |
| | NAtS | 0.4 | $79.5_{\pm4.5}$ | $74.4_{\pm6.3}$ | $88.4_{\pm3.6}$ | $89.8_{\pm2.2}$ |
| | | 0.6 | $64.5_{\pm5.5}$ | $60.9_{\pm4.1}$ | $74.5_{\pm3.7}$ | $74.6_{\pm2.5}$ |
| | | 0.8 | $53.3_{\pm2.4}$ | $51.9_{\pm1.6}$ | $56.7_{\pm2.3}$ | $56.3_{\pm1.8}$ |
| | ID | 1.0 | – | – | – | – |
| RSNA Bone Age | | 1.2 | $58.5_{\pm1.9}$ | $57.8_{\pm2.8}$ | $57.4_{\pm1.8}$ | $55.0_{\pm2.3}$ |
| | | 1.4 | $69.0_{\pm3.1}$ | $67.1_{\pm4.4}$ | $68.6_{\pm1.7}$ | $65.1_{\pm3.1}$ |
| | | 1.6 | $76.9_{\pm4.0}$ | $75.4_{\pm5.0}$ | $78.5_{\pm1.9}$ | $75.5_{\pm3.9}$ |
| | | 1.8 | $82.1_{\pm5.1}$ | $81.4_{\pm4.8}$ | $84.3_{\pm1.9}$ | $82.3_{\pm4.3}$ |
| | NAtS | 2.0 | $85.8_{\pm5.5}$ | $86.3_{\pm3.8}$ | $88.2_{\pm1.6}$ | $87.0_{\pm4.0}$ |
| | | 2.5 | $91.9_{\pm5.0}$ | $93.3_{\pm3.1}$ | $94.0_{\pm1.3}$ | $93.7_{\pm2.9}$ |
| | | 3.0 | $95.4_{\pm3.3}$ | $96.8_{\pm1.6}$ | $96.3_{\pm1.0}$ | $96.6_{\pm1.8}$ |
| | | 3.5 | $96.8_{\pm2.7}$ | $98.4_{\pm1.0}$ | $97.4_{\pm0.8}$ | $97.8_{\pm1.3}$ |
| | | 4.0 | $97.7_{\pm2.1}$ | $99.0_{\pm0.7}$ | $97.9_{\pm0.8}$ | $98.4_{\pm0.9}$ |
| | | 4.5 | $98.2_{\pm1.6}$ | $99.2_{\pm0.6}$ | $98.2_{\pm0.8}$ | $98.7_{\pm0.8}$ |

Table 4: NAtS detection performance on three NAtS datasets measured with Mahalanobis OOD detection method using AUROC.

# E    COMPARISON WITH ADDITIONAL BASELINES

| Dataset | CE | GODIN | Scaled Cosine | SSD+ | Ours |
|---|---|---|---|---|---|
| UKTFace | 94.6±06 | 94.7±0.8 | 94.4±0.7 | 94.2±0.5 | 94.1±0.7 |
| Amazon Review | 85.3±0.9 | 85.7±0.7 | 85.4±0.8 | 83.8±0.6 | 84.0±0.9 |
| RSNA Bone Age | 93.0±0.6 | 92.0±1.4 | 93.1±1.0 | 71.9±6.6 | 92.3±0.3 |

Table 5: Classification accuracy of the in-distribution in three NAtS datasets.

| | Distribution | Age | Baselines | | | | | Ours | |
|---|---|---|---|---|---|---|---|---|---|
| | | | Mahalanobis* | Gram matrix* | GODIN | Scaled Cosine | SSD+ | Mahalanobis | Mahalanobis* |
| | ID | 26 | – | – | – | – | – | – | – |
| | | 25 | 50.3±0.9 | 49.9±0.3 | 50.5±1.8 | 51.9±2.9 | 48.7±1.0 | 50.1±1.7 | 49.8±1.0 |
| | | 24 | 52.4±0.8 | 50.9±0.5 | 48.9±1.8 | 50.9±3.0 | 52.8±1.2 | 53.3±1.3 | 54.4±1.0 |
| | | 23 | 50.2±1.2 | 51.7±0.6 | 48.1±1.3 | 49.3±2.2 | 49.3±1.0 | 52.7±1.2 | 54.3±0.5 |
| | | 22 | 55.1±1.2 | 56.6±0.5 | 49.2±1.9 | 53.6±2.2 | 51.2±0.7 | 56.5±1.3 | 58.2±0.7 |
| | | 21 | 51.8±1.3 | 51.3±0.9 | 51.8±1.3 | 51.9±3.1 | 50.3±0.5 | 55.3±1.0 | 56.8±0.8 |
| UTKFace | | 19-20 | 55.7±1.0 | 56.6±0.3 | 47.9±1.2 | 54.8±1.8 | 56.7±0.7 | 57.9±1.2 | 58.8±0.3 |
| | | 17-18 | 56.9±1.2 | 61.3±0.7 | 46.6±2.0 | 58.1±2.7 | 58.2±1.0 | 61.0±1.1 | 61.8±0.6 |
| | NAtS | 15-16 | 65.7±1.2 | 63.3±0.5 | 45.9±1.9 | 66.5±2.5 | 67.1±1.8 | 70.7±0.7 | 70.7±1.5 |
| | | 12-14 | 61.6±1.1 | 67.7±0.7 | 45.1±2.8 | 69.5±4.3 | 71.5±1.4 | 75.3±0.9 | 75.9±1.9 |
| | | 9-11 | 63.7±1.0 | 70.8±1.2 | 43.8±2.2 | 71.3±5.7 | 73.4±1.6 | 80.5±0.9 | 80.8±1.4 |
| | | 7-8 | 62.8±1.4 | 73.1±1.0 | 45.0±1.6 | 69.4±7.3 | 72.5±1.4 | 82.1±1.9 | 82.6±1.3 |
| | | 5-6 | 61.8±1.3 | 70.6±0.5 | 44.1±3.0 | 68.9±5.9 | 74.7±1.4 | 83.5±1.4 | 83.0±1.8 |
| | | 3-4 | 68.4±1.0 | 75.3±0.6 | 44.3±1.7 | 68.1±9.4 | 80.8±1.3 | 88.6±1.0 | 89.4±1.2 |
| | | 2 | 63.5±1.0 | 74.4±1.1 | 44.2±1.9 | 65.4±10.9 | 79.5±1.2 | 88.3±1.6 | 88.9±1.3 |
| | | 1 | 65.8±1.9 | 80.3±0.9 | 43.3±2.7 | 63.2±10.4 | 79.7±1.8 | 90.3±2.0 | 91.5±1.1 |

| | Distribution | Year | Baselines | | | | | Ours | |
|---|---|---|---|---|---|---|---|---|---|
| | | | Mahalanobis* | Gram matrix* | GODIN | Scaled Cosine | SSD+ | Mahalanobis | Mahalanobis* |
| | ID | 2005 | – | – | – | – | – | – | – |
| | | 2006 | 51.4±0.0 | – | 50.5±0.1 | 49.4±0.5 | 51.3±0.3 | 51.4±0.1 | 51.4±0.0 |
| | | 2007 | 56.6±0.1 | – | 50.9±0.5 | 49.2±0.6 | 55.4±1.7 | 56.5±0.1 | 56.6±0.0 |
| Amazon Review | | 2008 | 54.9±0.0 | – | 50.8±0.1 | 49.4±0.5 | 53.8±0.2 | 54.9±0.1 | 54.8±0.1 |
| | | 2009 | 53.6±0.1 | – | 50.9±0.3 | 49.3±0.6 | 52.5±1.2 | 53.6±0.1 | 53.6±0.1 |
| | NAtS | 2010 | 53.6±0.1 | – | 50.5±0.3 | 48.8±0.6 | 52.6±1.3 | 53.6±0.1 | 53.6±0.1 |
| | | 2011 | 55.0±0.1 | – | 50.5±0.5 | 47.4±0.7 | 53.0±1.5 | 54.9±0.0 | 54.8±0.0 |
| | | 2012 | 62.4±0.1 | – | 50.8±0.6 | 45.9±1.3 | 58.2±2.6 | 62.6±0.1 | 62.5±0.1 |
| | | 2013 | 75.0±0.1 | – | 51.2±1.1 | 45.2±1.8 | 67.4±7.7 | 75.1±0.0 | 75.0±0.1 |
| | | 2014 | 76.5±0.1 | – | 51.1±0.6 | 45.8±1.6 | 69.1±7.2 | 76.7±0.1 | 76.7±0.1 |

| | Distribution | Brightness | Baselines | | | | | Ours | |
|---|---|---|---|---|---|---|---|---|---|
| | | | Mahalanobis* | Gram matrix* | GODIN | Scaled Cosine | SSD+ | Mahalanobis | Mahalanobis* |
| | | 0.0 | 100.0±0.1 | 100.0±0.0 | 97.8±0.6 | 97.5±2.1 | 100.0±0.0 | 100.0±0.0 | 100.0±0.0 |
| | | 0.2 | 93.9±2.5 | 100.0±0.0 | 48.9±1.0 | 72.9±9.3 | 96.1±5.9 | 98.0±1.0 | 89.3±12.3 |
| | NAtS | 0.4 | 77.8±9.6 | 99.0±0.3 | 51.8±1.2 | 59.9±3.4 | 92.1±7.9 | 89.8±2.2 | 67.2±18.2 |
| | | 0.6 | 65.2±8.9 | 61.9±3.5 | 50.7±1.6 | 54.1±3.2 | 81.9±7.4 | 74.6±2.5 | 58.4±12.9 |
| | | 0.8 | 54.2±3.6 | 37.9±1.5 | 51.1±1.3 | 50.8±2.2 | 62.3±3.6 | 56.3±1.8 | 52.0±6.1 |
| | ID | 1.0 | – | – | – | – | – | – | – |
| RSNA Bone Age | | 1.2 | 65.8±1.8 | 58.6±0.9 | 48.9±1.0 | 53.5±2.6 | 66.6±3.2 | 55.0±2.4 | 61.2±1.5 |
| | | 1.4 | 79.3±1.9 | 63.5±1.0 | 49.2±0.6 | 58.4±4.5 | 83.8±5.1 | 65.1±3.1 | 73.4±2.6 |
| | | 1.6 | 88.4±2.4 | 69.1±1.0 | 49.1±1.4 | 62.9±5.6 | 92.7±5.4 | 75.5±3.9 | 83.1±5.1 |
| | | 1.8 | 93.5±1.9 | 74.9±1.1 | 49.7±1.4 | 66.6±7.3 | 96.2±4.4 | 82.3±4.3 | 89.4±6.0 |
| | NAtS | 2.0 | 96.6±1.1 | 80.8±1.6 | 50.1±0.7 | 69±10.2 | 97.6±3.2 | 87.0±4.0 | 93.2±5.4 |
| | | 2.5 | 99.5±0.2 | 91.3±1.4 | 48.2±2.0 | 75.4±13.4 | 99.3±1.1 | 93.7±2.9 | 98.1±2.6 |
| | | 3.0 | 99.9±0.1 | 95.7±1.0 | 49.1±2.7 | 83.0±9.4 | 99.8±0.5 | 96.6±1.8 | 99.3±1.0 |
| | | 3.5 | 99.9±0.1 | 97.6±0.5 | 50.1±3.0 | 87.3±5.2 | 99.9±0.2 | 97.8±1.3 | 99.7±0.3 |
| | | 4.0 | 100.0±0.1 | 98.4±0.4 | 51.2±3.5 | 89.3±2.7 | 100.0±0.1 | 98.4±0.9 | 99.9±0.1 |
| | | 4.5 | 100.0±0.0 | 98.9±0.2 | 51.6±3.3 | 90.2±2.5 | 100.0±0.1 | 98.7±0.8 | 99.9±0.1 |

Table 6: NAtS detection performance on distributional shifts in three NAtS datasets measured by AUROC. Mahalanobis and Mahalanobis* indicate using penultimate layer only or ensembling all layer respectively. Gram matrix* also denotes the ensembled version.

In this section, we investigate NAtS detection performance using additional baselines, including recent work, and compare them with those of the model trained with our modified loss on three NAtS datasets. Firstly, as we mentioned in the main paper, we evaluate the model trained with cross-entropy loss with ensembled version of Mahalanobis detector (Lee et al., 2018) and the method that utilizes the gram matrix in OOD detection (Chandramouli & Sageev, 2020).

We also consider the other recent baselines for the comparison. Recently, Sehwag et al. (2021) propose a unified framework to leverage self-supervised learning for OOD detection. We set the

method as our baseline and denote this baseline as SSD$^+$. The experiment on SSD$^+$ requires a data augmentation strategy. In the Image domain, we augment the training dataset as Chen et al. (2020). We construct an augmented training dataset using a back-translation model for text experiments as (Fang et al., 2020; Gunel et al., 2020). We train SSD$^+$ for 100 epochs for UTKFace and RSNA Bone age dataset and 50 epochs for Amazon Review dataset.

Some recent methods (Hsu et al., 2020; Techapanurak et al., 2020) exploit the cosine similarity to detect OOD samples. Hsu et al. (2020) suggest to decompose confidence scores and to modify input preprocessing method in ODIN (Liang et al., 2018). For OOD detection, they calculate the cosine similarity between the features from the penultimate layer and the class specific weights and use the maximum value as an OOD score. Techapanurak et al. (2020) also propose to compute cosine similarity between the class weight vector and the penultimate layer feature of each input to obtain OOD scores. Specifically, they train the model with cross entropy loss as standard classification model, but logit value is set by the scaled cosine similarity value. After training, the cosine similarity values between the penultimate layer feature of the input and the weight vector for each class are calculated. The maximum cosine similarity value is used as the OOD score for the samples. We name the methods as GODIN and Scaled Cosine in Table 6, respectively.

NAtS detection performance of our method and other baselines are provided in Table 6. In UTKFace dataset, GODIN, Scaled Cosine, and ensembled version of Mahalanobis detector have relatively low detection performance than the other baselines. Although Gram matrix shows reasonable detection performance in UTKFace and RSNA Bone Age NAtS groups, this method is hard to applied to the Amazon Book Review dataset because the notion of gram matrix is vague in the text domain. In three NAtS categories, SSD+ has the most robust detection performance among the five baselines, however, our simple remedy in training loss help Mahalanobis detector to demonstrate improved performance particularly under the UTKFace NAtS dataset. In conclusion, when compared with the five more recent baselines, our method shows robust and high detection performance in all the three NAtS datasets, regardless of the domains. Scaled Cosine, which compute logit based on cosine similarity, have similar tendency with confidence-based methods.

## F  QUANTITATIVE RESULTS WITH OTHER METRICS

We report the NAtS detection performance of baselines and our method based on four other metrics: Detection Accuracy, AUPR-In, AUPR-Out, and TNR@95%TPR, which are often used in the OOD detection community. We present the detection performance on NAtS datasets in three domain measured by Detection Accuracy, AUPR-In, AUPR-Out, and TNR@95%TPR in Tables 7,8,9, and 10 respectively. The results indicate that our simple modification in the training objective improves the performance of the Mahalanobis detector, thus making it a robust NAtS detection method for the three NAtS categories presented in the paper. Specifically, in UTKFace, we observe that using our suggested training loss results in a significant increase in NAtS detection performance using the Mahalanobis detector. At the same time, the suggested method effectively detects the NAtS samples in the other two datasets. In summary, based on the results obtained using five different metrics (AUROC, Detection Accuracy, AUPR-In, AUPR-Out, and TNR@95%TPR), our suggested modification makes the Mahalanobis detector a general NAtS detection method for all three NAtS categories.

| Distribution | Age | CE | | | | | Ours |
| | | MSP | ODIN | Mahalanobis | Gram matrix | Energy score | Mahalanobis |
|---|---|---|---|---|---|---|---|
| ID | 26 | – | – | – | – | – | – |
| NAtS | 25 | 6.3±3.1 | 6.1±1.6 | 5.5±1.5 | 5.5±1.8 | 6.0±2.5 | 7.1±2.6 |
| | 24 | 6.0±1.5 | 4.1±0.8 | 5.1±1.7 | 6.6±1.5 | 5.5±2.0 | 4.6±1.2 |
| | 23 | 5.3±1.9 | 3.8±0.9 | 4.6±1.2 | 7.0±1.9 | 5.4±2.2 | 5.3±1.5 |
| | 22 | 7.2±2.2 | 4.5±0.9 | 5.0±1.7 | 6.9±2.0 | 6.7±1.7 | 8.6±2.6 |
| | 21 | 5.7±3.1 | 5.7±1.2 | 6.0±1.9 | 6.2±1.3 | 5.9±2.7 | 7.9±1.8 |
| | 19-20 | 6.7±2.3 | 4.8±1.2 | 4.8±0.9 | 7.6±2.5 | 6.2±1.5 | 8.1±2.0 |
| | 17-18 | 9.8±2.4 | 7.5±0.4 | 5.2±1.8 | 8.9±2.1 | 10.1±1.9 | 11.1±3.3 |
| | 15-16 | 15.2±5.7 | 9.4±2.1 | 6.7±2.1 | 11.7±2.0 | 14.2±4.5 | 17.1±4.7 |
| | 12-14 | 19.2±4.9 | 17.7±3.5 | 4.5±1.9 | 10.9±3.5 | 17.2±5.2 | 21.7±4.7 |
| | 9-11 | 19.1±6.8 | 23.9±3.6 | 4.3±1.7 | 10.7±3.2 | 18.4±5.9 | 26.5±5.5 |
| | 7-8 | 17.9±4.5 | 27.4±5.3 | 4.9±1.7 | 10.4±4.3 | 18.2±5.7 | 29.3±7.4 |
| | 5-6 | 19.2±6.0 | 28.5±3.9 | 5.6±0.9 | 10.0±2.7 | 18.4±6.0 | 29.8±9.5 |
| | 3-4 | 16.3±5.0 | 31.9±3.1 | 6.7±2.2 | 12.5±3.2 | 13.5±3.1 | 41.5±8.8 |
| | 2 | 10.6±3.5 | 37.6±6.2 | 4.9±1.6 | 9.0±2.2 | 10.9±3.3 | 39.4±11.6 |
| | 1 | 12.7±4.1 | 47.5±2.7 | 4.9±2.6 | 10.8±4.1 | 10.5±3.1 | 46.0±15.6 |

| Distribution | Year | CE | | | | | Ours |
| | | MSP | ODIN | Mahalanobis | Gram matrix | Energy score | Mahalanobis |
|---|---|---|---|---|---|---|---|
| ID | 2005 | – | – | – | – | – | – |
| NAtS | 2006 | 5.1±0.6 | 5.1±0.5 | 5.4±0.3 | – | 4.8±0.4 | 5.4±0.2 |
| | 2007 | 4.8±0.4 | 5.1±0.5 | 8.2±0.3 | – | 4.8±0.5 | 8.5±0.3 |
| | 2008 | 4.4±0.4 | 4.9±0.8 | 8.1±0.3 | – | 4.5±0.7 | 8.7±0.2 |
| | 2009 | 5.0±1.0 | 4.7±0.5 | 7.7±0.3 | – | 4.9±0.9 | 8.4±0.4 |
| | 2010 | 4.7±0.5 | 4.9±0.8 | 7.8±0.2 | – | 4.8±0.7 | 8.1±0.2 |
| | 2011 | 4.4±0.5 | 4.4±0.6 | 7.4±0.4 | – | 4.6±0.7 | 8.3±0.2 |
| | 2012 | 3.7±0.4 | 4.7±0.7 | 14.2±0.6 | – | 4.1±1.1 | 16.4±0.5 |
| | 2013 | 3.7±0.2 | 4.5±0.7 | 25.9±0.8 | – | 4.5±1.5 | 30.9±0.6 |
| | 2014 | 3.8±0.5 | 5.4±0.8 | 25.8±0.9 | – | 4.3±1.3 | 30.2±0.7 |

| Distribution | Brightness | CE | | | | | Ours |
| | | MSP | ODIN | Mahalanobis | Gram matrix | Energy score | Mahalanobis |
|---|---|---|---|---|---|---|---|
| NAtS | 0.0 | 40.0±54.8 | 100.0±0.0 | 100.0±0.0 | 100.0±0.0 | 40.0±54.8 | 100.0±0.0 |
| | 0.2 | 12.1±4.0 | 59.3±14.5 | 50.2±18.8 | 17.4±6.2 | 10.3±6.3 | 93.6±5.8 |
| | 0.4 | 7.2±2.7 | 13.2±3.3 | 15.4±8.9 | 7.6±2.2 | 6.1±3.3 | 36.0±3.7 |
| | 0.6 | 5.3±2.0 | 4.8±1.3 | 8.7±4.3 | 5.1±1.3 | 4.9±2.6 | 14.4±1.8 |
| | 0.8 | 5.1±1.4 | 4.2±1.1 | 5.9±1.5 | 5.0±0.8 | 4.4±1.9 | 6.7±1.9 |
| ID | 1.0 | – | – | – | – | – | – |
| NAtS | 1.2 | 5.6±1.9 | 8.4±0.8 | 7.9±1.6 | 5.7±1.2 | 5.5±2.1 | 6.7±1.1 |
| | 1.4 | 7.3±1.5 | 18.1±3.5 | 13.0±4.3 | 9.3±2.5 | 7.4±1.9 | 13.4±2.6 |
| | 1.6 | 10.2±3.5 | 29.3±4.9 | 20.8±5.9 | 13.3±2.9 | 10.5±3.1 | 22.8±4.0 |
| | 1.8 | 11.5±3.5 | 41.8±7.2 | 29.0±9.6 | 16.9±4.0 | 10.9±2.5 | 29.7±7.4 |
| | 2.0 | 11.5±3.7 | 55.2±9.0 | 36.0±12.2 | 18.2±4.6 | 11.1±2.7 | 37.8±11.9 |
| | 2.5 | 11.5±2.1 | 77.7±9.6 | 53.3±20.4 | 24.4±6.6 | 10.4±2.1 | 61.6±13.9 |
| | 3.0 | 13.3±3.1 | 92.1±5.3 | 72.3±23.2 | 31.7±9.7 | 11.9±3.2 | 82.5±10.6 |
| | 3.5 | 15.5±5.0 | 97.5±1.9 | 81.5±24.8 | 37.7±10.8 | 13.8±2.8 | 91.8±7.6 |
| | 4.0 | 16.8±6.1 | 99.3±0.8 | 85.6±22.9 | 44.0±11.6 | 14.6±4.1 | 96.1±3.8 |
| | 4.5 | 18.9±7.8 | 99.6±0.5 | 89.9±18.3 | 50.2±13.8 | 16.0±5.4 | 97.9±2.6 |

Table 7: NAtS detection performance on three NAtS datasets measured by TNR at TPR 95%.

| | Distribution | Age | CE | | | | | Ours |
|---|---|---|---|---|---|---|---|---|
| | | | MSP | ODIN | Mahalanobis | Gram matrix | Energy score | Mahalanobis |
| UTKFace | ID | 26 | – | – | – | – | – | – |
| | NAtS | 25 | $53.1_{\pm0.6}$ | $51.6_{\pm0.8}$ | $53.0_{\pm0.9}$ | $52.4_{\pm0.8}$ | $52.8_{\pm0.4}$ | $53.4_{\pm1.7}$ |
| | | 24 | $53.9_{\pm1.0}$ | $53.6_{\pm1.0}$ | $53.7_{\pm1.2}$ | $52.8_{\pm0.7}$ | $54.0_{\pm0.9}$ | $54.9_{\pm1.0}$ |
| | | 23 | $53.4_{\pm0.5}$ | $52.3_{\pm0.2}$ | $52.3_{\pm1.7}$ | $53.0_{\pm0.7}$ | $53.2_{\pm0.9}$ | $54.8_{\pm0.5}$ |
| | | 22 | $53.7_{\pm0.8}$ | $52.8_{\pm1.5}$ | $53.4_{\pm1.5}$ | $53.7_{\pm1.4}$ | $54.0_{\pm1.0}$ | $57.3_{\pm1.5}$ |
| | | 21 | $55.2_{\pm1.3}$ | $53.8_{\pm1.2}$ | $52.8_{\pm1.8}$ | $53.0_{\pm1.1}$ | $55.2_{\pm1.1}$ | $55.6_{\pm0.8}$ |
| | | 19-20 | $56.5_{\pm1.3}$ | $53.9_{\pm1.0}$ | $54.6_{\pm1.7}$ | $53.2_{\pm0.9}$ | $56.2_{\pm0.9}$ | $57.2_{\pm0.7}$ |
| | | 17-18 | $60.8_{\pm1.3}$ | $56.7_{\pm0.5}$ | $55.6_{\pm1.8}$ | $54.4_{\pm0.7}$ | $60.2_{\pm1.0}$ | $59.3_{\pm0.6}$ |
| | | 15-16 | $66.3_{\pm1.3}$ | $59.5_{\pm0.8}$ | $59.4_{\pm2.0}$ | $56.9_{\pm1.3}$ | $65.8_{\pm1.5}$ | $66.4_{\pm1.0}$ |
| | | 12-14 | $70.4_{\pm1.3}$ | $65.1_{\pm1.4}$ | $58.2_{\pm2.3}$ | $57.8_{\pm1.8}$ | $70.0_{\pm1.1}$ | $69.2_{\pm0.5}$ |
| | | 9-11 | $72.5_{\pm1.4}$ | $71.6_{\pm1.9}$ | $57.2_{\pm2.9}$ | $56.2_{\pm2.7}$ | $72.1_{\pm1.4}$ | $74.2_{\pm0.9}$ |
| | | 7-8 | $72.8_{\pm1.0}$ | $72.5_{\pm1.6}$ | $55.4_{\pm2.9}$ | $56.4_{\pm1.6}$ | $72.6_{\pm1.4}$ | $75.2_{\pm1.6}$ |
| | | 5-6 | $74.0_{\pm1.6}$ | $74.8_{\pm2.0}$ | $54.8_{\pm2.2}$ | $55.8_{\pm2.4}$ | $73.8_{\pm1.8}$ | $76.2_{\pm1.1}$ |
| | | 3-4 | $75.6_{\pm1.2}$ | $80.5_{\pm2.5}$ | $57.2_{\pm2.2}$ | $56.0_{\pm0.7}$ | $74.9_{\pm1.6}$ | $81.5_{\pm0.4}$ |
| | | 2 | $76.0_{\pm1.1}$ | $83.3_{\pm2.1}$ | $55.9_{\pm1.0}$ | $54.1_{\pm0.3}$ | $75.2_{\pm1.8}$ | $81.6_{\pm1.4}$ |
| | | 1 | $76.9_{\pm1.0}$ | $84.6_{\pm1.2}$ | $54.0_{\pm2.6}$ | $54.1_{\pm2.0}$ | $76.2_{\pm1.7}$ | $83.0_{\pm1.7}$ |

| | Distribution | Year | CE | | | | | Ours |
|---|---|---|---|---|---|---|---|---|
| | | | MSP | ODIN | Mahalanobis | Gram matrix | Energy score | Mahalanobis |
| Amazon Review | ID | 2005 | – | – | – | – | – | – |
| | NAtS | 2006 | $50.6_{\pm0.1}$ | $50.6_{\pm0.1}$ | $51.7_{\pm0.1}$ | – | $50.5_{\pm0.2}$ | $51.6_{\pm0.1}$ |
| | | 2007 | $50.2_{\pm0.1}$ | $50.1_{\pm0.2}$ | $54.9_{\pm0.1}$ | – | $50.2_{\pm0.1}$ | $55.0_{\pm0.1}$ |
| | | 2008 | $50.1_{\pm0.1}$ | $50.3_{\pm0.2}$ | $54.4_{\pm0.2}$ | – | $50.2_{\pm0.2}$ | $54.5_{\pm0.2}$ |
| | | 2009 | $50.4_{\pm0.4}$ | $50.4_{\pm0.2}$ | $53.6_{\pm0.1}$ | – | $50.3_{\pm0.3}$ | $53.6_{\pm0.1}$ |
| | | 2010 | $50.2_{\pm0.1}$ | $50.3_{\pm0.3}$ | $53.3_{\pm0.1}$ | – | $50.2_{\pm0.1}$ | $53.4_{\pm0.1}$ |
| | | 2011 | $50.2_{\pm0.1}$ | $50.1_{\pm0.1}$ | $54.5_{\pm0.1}$ | – | $50.3_{\pm0.2}$ | $54.6_{\pm0.1}$ |
| | | 2012 | $50.0_{\pm0.0}$ | $50.2_{\pm0.2}$ | $60.3_{\pm0.1}$ | – | $50.3_{\pm0.3}$ | $60.4_{\pm0.1}$ |
| | | 2013 | $50.0_{\pm0.0}$ | $50.2_{\pm0.1}$ | $70.3_{\pm0.2}$ | – | $50.4_{\pm0.4}$ | $70.4_{\pm0.2}$ |
| | | 2014 | $50.1_{\pm0.1}$ | $50.5_{\pm0.3}$ | $70.2_{\pm0.1}$ | – | $50.4_{\pm0.3}$ | $70.2_{\pm0.1}$ |

| | Distribution | Brightness | CE | | | | | Ours |
|---|---|---|---|---|---|---|---|---|
| | | | MSP | ODIN | Mahalanobis | Gram matrix | Energy score | Mahalanobis |
| RSNA Bone Age | NAtS | 0.0 | $97.0_{\pm2.1}$ | $100.0_{\pm0.0}$ | $99.9_{\pm0.1}$ | $99.9_{\pm0.2}$ | $96.1_{\pm2.1}$ | $100.0_{\pm0.0}$ |
| | | 0.2 | $68.3_{\pm2.8}$ | $82.8_{\pm6.3}$ | $89.6_{\pm1.9}$ | $68.5_{\pm3.2}$ | $68.0_{\pm3.4}$ | $96.1_{\pm1.2}$ |
| | | 0.4 | $55.2_{\pm1.2}$ | $62.1_{\pm3.0}$ | $74.9_{\pm4.4}$ | $54.8_{\pm3.2}$ | $55.3_{\pm1.6}$ | $84.5_{\pm2.6}$ |
| | | 0.6 | $52.9_{\pm1.1}$ | $54.0_{\pm1.6}$ | $61.9_{\pm4.4}$ | $52.1_{\pm1.7}$ | $53.0_{\pm1.0}$ | $70.5_{\pm3.0}$ |
| | | 0.8 | $51.5_{\pm0.6}$ | $51.1_{\pm0.9}$ | $53.9_{\pm1.7}$ | $51.0_{\pm0.4}$ | $51.6_{\pm0.5}$ | $56.0_{\pm1.5}$ |
| | ID | 1.0 | – | – | – | – | – | – |
| | NAtS | 1.2 | $53.2_{\pm0.7}$ | $54.4_{\pm0.9}$ | $57.3_{\pm1.1}$ | $52.1_{\pm0.9}$ | $53.2_{\pm0.8}$ | $55.5_{\pm2.2}$ |
| | | 1.4 | $55.8_{\pm0.8}$ | $60.4_{\pm2.1}$ | $65.1_{\pm2.2}$ | $55.2_{\pm1.1}$ | $55.4_{\pm0.8}$ | $62.5_{\pm2.7}$ |
| | | 1.6 | $58.6_{\pm0.9}$ | $66.3_{\pm3.0}$ | $71.1_{\pm3.0}$ | $59.7_{\pm1.8}$ | $58.7_{\pm0.9}$ | $70.8_{\pm3.0}$ |
| | | 1.8 | $60.8_{\pm1.1}$ | $73.1_{\pm4.3}$ | $75.7_{\pm4.4}$ | $62.4_{\pm2.0}$ | $60.7_{\pm1.1}$ | $76.5_{\pm2.6}$ |
| | | 2.0 | $61.2_{\pm1.0}$ | $79.4_{\pm4.7}$ | $79.2_{\pm5.0}$ | $63.9_{\pm3.5}$ | $61.2_{\pm1.1}$ | $80.8_{\pm3.0}$ |
| | | 2.5 | $63.4_{\pm3.3}$ | $88.5_{\pm3.7}$ | $86.6_{\pm5.2}$ | $68.4_{\pm4.5}$ | $63.1_{\pm2.9}$ | $87.7_{\pm3.9}$ |
| | | 3.0 | $70.4_{\pm2.9}$ | $94.3_{\pm2.5}$ | $91.4_{\pm4.0}$ | $74.8_{\pm3.8}$ | $70.3_{\pm2.7}$ | $92.3_{\pm2.8}$ |
| | | 3.5 | $75.8_{\pm1.7}$ | $96.9_{\pm1.4}$ | $94.0_{\pm3.4}$ | $78.4_{\pm4.9}$ | $75.5_{\pm1.7}$ | $94.8_{\pm2.3}$ |
| | | 4.0 | $78.9_{\pm1.8}$ | $98.3_{\pm0.9}$ | $95.4_{\pm2.3}$ | $82.1_{\pm5.4}$ | $78.6_{\pm1.7}$ | $96.4_{\pm1.7}$ |
| | | 4.5 | $80.9_{\pm2.5}$ | $98.8_{\pm0.8}$ | $96.0_{\pm2.0}$ | $85.0_{\pm4.4}$ | $80.7_{\pm2.4}$ | $97.1_{\pm1.6}$ |

Table 8: NAtS detection performance on three NAtS datasets measured by detection accuracy.

| | Distribution | Age | CE | | | | | Ours |
|---|---|---|---|---|---|---|---|---|
| | | | MSP | ODIN | Mahalanobis | Gram matrix | Energy score | Mahalanobis |
| UTKFace | ID | 26 | – | – | – | – | – | – |
| | NAtS | 25 | 48.2±1.4 | 46.3±0.5 | 49.0±0.6 | 31.9±0.5 | 48.1±1.3 | 48.5±1.8 |
| | | 24 | 51.7±1.2 | 50.6±0.7 | 50.1±1.7 | 32.2±0.4 | 51.5±1.1 | 51.9±1.9 |
| | | 23 | 51.2±1.0 | 49.7±0.5 | 47.2±1.2 | 32.3±0.6 | 50.8±0.9 | 51.2±1.4 |
| | | 22 | 51.7±1.0 | 49.2±0.7 | 49.7±1.0 | 32.7±0.5 | 51.6±0.8 | 53.7±2.4 |
| | | 21 | 54.7±1.4 | 53.7±1.5 | 48.7±1.3 | 32.4±0.7 | 54.5±1.4 | 53.9±0.9 |
| | | 19-20 | 56.3±1.6 | 53.1±0.7 | 51.2±0.9 | 32.3±0.4 | 56.0±1.4 | 56.5±2.4 |
| | | 17-18 | 61.8±1.8 | 57.5±1.1 | 51.5±0.9 | 33.0±0.8 | 61.4±1.8 | 58.5±1.8 |
| | | 15-16 | 69.2±1.3 | 63.1±1.7 | 60.0±1.9 | 34.3±0.7 | 68.5±1.8 | 70.2±0.9 |
| | | 12-14 | 77.2±1.5 | 70.5±1.8 | 57.8±2.9 | 34.9±1.2 | 76.5±1.7 | 76.4±0.9 |
| | | 9-11 | 79.2±1.1 | 78.6±2.3 | 55.1±2.9 | 34.0±1.6 | 78.9±2.3 | 81.6±1.3 |
| | | 7-8 | 78.7±0.8 | 80.2±2.0 | 53.6±2.7 | 34.1±1.2 | 78.2±2.0 | 83.3±1.3 |
| | | 5-6 | 82.1±1.1 | 81.7±2.4 | 52.1±1.8 | 33.7±1.5 | 81.9±1.5 | 85.1±1.0 |
| | | 3-4 | 83.7±1.4 | 87.4±2.6 | 56.6±3.3 | 33.6±0.8 | 83.0±2.1 | 89.9±0.7 |
| | | 2 | 82.7±0.5 | 88.7±1.6 | 54.5±3.2 | 32.8±0.4 | 82.3±1.3 | 88.9±0.9 |
| | | 1 | 85.2±0.6 | 92.0±0.9 | 53.0±4.4 | 32.5±1.1 | 84.6±1.4 | 91.6±1.3 |

| | Distribution | Year | CE | | | | | Ours |
|---|---|---|---|---|---|---|---|---|
| | | | MSP | ODIN | Mahalanobis | Gram matrix | Energy score | Mahalanobis |
| Amazon Review | ID | 2005 | – | – | – | – | – | – |
| | NAtS | 2006 | 49.9±0.2 | 49.9±0.4 | 50.7±0.1 | – | 49.8±0.3 | 50.9±0.1 |
| | | 2007 | 47.9±0.3 | 48.4±0.6 | 54.3±0.1 | – | 47.9±0.4 | 54.6±0.1 |
| | | 2008 | 47.8±0.3 | 48.1±0.7 | 52.7±0.1 | – | 47.8±0.3 | 52.9±0.1 |
| | | 2009 | 48.0±0.2 | 48.5±0.5 | 51.7±0.1 | – | 48.1±0.3 | 51.9±0.1 |
| | | 2010 | 48.5±0.2 | 49.0±0.6 | 51.9±0.1 | – | 48.5±0.3 | 52.2±0.1 |
| | | 2011 | 46.9±0.2 | 47.7±0.8 | 52.9±0.1 | – | 47.0±0.2 | 53.2±0.1 |
| | | 2012 | 43.6±0.4 | 45.1±1.0 | 57.9±0.1 | – | 43.8±0.4 | 58.2±0.1 |
| | | 2013 | 40.6±0.3 | 42.4±1.1 | 68.8±0.1 | – | 40.8±0.4 | 70.3±0.1 |
| | | 2014 | 41.2±0.3 | 42.7±1.1 | 72.3±0.3 | – | 41.3±0.4 | 74.3±0.2 |

| | Distribution | Brightness | CE | | | | | Ours |
|---|---|---|---|---|---|---|---|---|
| | | | MSP | ODIN | Mahalanobis | Gram matrix | Energy score | Mahalanobis |
| RSNA Bone Age | NAtS | 0.0 | 96.8±2.1 | 99.9±0.0 | 99.8±0.1 | 99.7±0.3 | 95.9±2.1 | 99.9±0.0 |
| | | 0.2 | 73.1±4.4 | 88.8±6.4 | 95.4±1.3 | 43.5±2.0 | 72.2±5.1 | 98.5±0.6 |
| | | 0.4 | 55.7±2.7 | 64.5±5.4 | 81.8±4.6 | 34.4±2.0 | 54.8±3.5 | 92.2±1.9 |
| | | 0.6 | 52.3±1.7 | 53.3±2.9 | 65±6.4 | 32.8±1.1 | 52.2±1.6 | 77.4±2.9 |
| | | 0.8 | 49.9±1.0 | 49.1±1.2 | 53.4±2.9 | 32.2±0.6 | 50.0±0.8 | 56.6±1.6 |
| | ID | 1.0 | – | – | – | – | – | – |
| | NAtS | 1.2 | 51.0±1.8 | 54.2±1.7 | 58.5±2.6 | 32.8±1.0 | 50.6±1.6 | 54.8±1.4 |
| | | 1.4 | 53.0±3.5 | 61.5±3.3 | 69.4±4.1 | 34.4±0.9 | 52.2±3.1 | 65.8±2.3 |
| | | 1.6 | 55.5±5.5 | 69.3±4.9 | 77.5±4.7 | 37.0±1.4 | 54.4±4.7 | 77.5±3.3 |
| | | 1.8 | 58.1±6.6 | 78.3±5.8 | 83.0±5.7 | 38.7±1.6 | 56.9±5.5 | 84.8±3.3 |
| | | 2.0 | 60.6±7.0 | 86.2±5.0 | 87.0±5.8 | 39.8±2.5 | 59.4±6.4 | 89.2±3.0 |
| | | 2.5 | 66.4±7.8 | 94.9±2.4 | 92.8±5.1 | 43.8±3.9 | 65.4±8.2 | 94.8±2.4 |
| | | 3.0 | 77.3±5.4 | 98.2±0.9 | 96.1±2.8 | 50.5±4.3 | 76.6±5.9 | 97.2±1.5 |
| | | 3.5 | 83.5±2.8 | 99.2±0.4 | 97.5±2.0 | 55.7±7.4 | 83.0±2.9 | 98.3±1.0 |
| | | 4.0 | 87.3±1.8 | 99.5±0.2 | 98.2±1.4 | 63.5±10.9 | 86.9±1.5 | 98.8±0.7 |
| | | 4.5 | 89.0±2.4 | 99.6±0.2 | 98.6±1.1 | 69.4±11.0 | 88.6±2.1 | 99.0±0.5 |

Table 9: NAtS detection performance on three NAtS datasets measured by AUPR-In.

| | Distribution | Age | CE | | | | | Ours |
|---|---|---|---|---|---|---|---|---|
| | | | MSP | ODIN | Mahalanobis | Gram matrix | Energy score | Mahalanobis |
| UTKFace | ID | 26 | – | – | – | – | – | – |
| | NAtS | 25 | 51.7±1.9 | 49.3±1.2 | 51.2±0.9 | 41.3±1.2 | 51.2±1.5 | 52.2±2.1 |
| | | 24 | 52.5±1.6 | 50.1±0.5 | 50.9±2.7 | 42.5±1.2 | 52.3±1.2 | 52.9±1.5 |
| | | 23 | 50.9±1.8 | 47.9±0.5 | 49.6±2.3 | 42.8±1.4 | 50.2±1.1 | 52.4±1.7 |
| | | 22 | 53.4±2.0 | 49.3±0.9 | 51.9±2.4 | 43.6±1.5 | 52.9±1.2 | 56.9±1.1 |
| | | 21 | 53.5±1.9 | 50.6±0.6 | 51.3±2.8 | 42.6±1.2 | 53.0±1.7 | 55.5±1.4 |
| | | 19-20 | 55.2±2.1 | 50.5±0.9 | 52.7±1.3 | 43.1±1.2 | 54.7±1.3 | 57.5±1.5 |
| | | 17-18 | 60.7±1.9 | 55.2±1.0 | 53.0±1.8 | 45.4±1.7 | 60.5±0.9 | 60.9±2.0 |
| | | 15-16 | 67.8±2.6 | 58.0±2.0 | 56.9±2.8 | 49.1±1.5 | 67.2±1.4 | 69.5±0.5 |
| | | 12-14 | 73.2±2.5 | 67.2±2.2 | 54.5±2.4 | 49.5±2.1 | 72.3±2.0 | 73.4±1.2 |
| | | 9-11 | 74.2±2.6 | 74.6±2.8 | 53.8±3.8 | 47.6±4.0 | 73.6±2.2 | 78.2±1.8 |
| | | 7-8 | 74.0±2.2 | 76.1±2.4 | 52.1±4.5 | 48.2±3.2 | 73.4±2.1 | 79.9±2.7 |
| | | 5-6 | 75.2±2.3 | 77.7±3.0 | 52.7±2.3 | 47.2±3.2 | 74.4±2.1 | 80.7±3.0 |
| | | 3-4 | 73.9±2.4 | 81.4±2.1 | 54.7±3.2 | 48.1±1.6 | 72.7±1.6 | 86.5±2.1 |
| | | 2 | 71.8±1.6 | 84.6±2.2 | 52.1±2.8 | 45.0±0.9 | 71.0±2.7 | 85.8±3.5 |
| | | 1 | 73.3±1.7 | 88.0±1.4 | 51.1±4.1 | 44.7±4.0 | 71.8±2.2 | 88.2±3.4 |

| | Distribution | Year | CE | | | | | Ours |
|---|---|---|---|---|---|---|---|---|
| | | | MSP | ODIN | Mahalanobis | Gram matrix | Energy score | Mahalanobis |
| Amazon Review | ID | 2005 | – | – | – | – | – | – |
| | NAtS | 2006 | 49.8±0.5 | 49.9±0.3 | 51.4±0.0 | – | 49.7±0.3 | 51.4±0.1 |
| | | 2007 | 48.9±0.4 | 49.4±0.6 | 56.0±0.0 | – | 48.8±0.7 | 56.4±0.1 |
| | | 2008 | 47.9±0.5 | 48.4±0.7 | 54.4±0.0 | – | 48.0±0.5 | 54.9±0.1 |
| | | 2009 | 48.7±0.7 | 48.9±0.2 | 53.7±0.0 | – | 48.6±0.6 | 54.1±0.0 |
| | | 2010 | 48.2±0.4 | 48.9±0.8 | 53.6±0.0 | – | 48.4±0.7 | 54.0±0.1 |
| | | 2011 | 46.9±0.4 | 47.3±0.8 | 53.9±0.0 | – | 47.1±0.6 | 54.5±0.0 |
| | | 2012 | 44.8±0.6 | 46.8±1.2 | 62.2±0.0 | – | 45.4±1.2 | 63.4±0.1 |
| | | 2013 | 43.6±0.5 | 46.0±1.0 | 73.9±0.1 | – | 44.4±1.3 | 75.9±0.1 |
| | | 2014 | 44.0±0.7 | 46.6±1.1 | 74.4±0.1 | – | 44.5±1.2 | 76.2±0.1 |

| | Distribution | Brightness | CE | | | | | Ours |
|---|---|---|---|---|---|---|---|---|
| | | | MSP | ODIN | Mahalanobis | Gram matrix | Energy score | Mahalanobis |
| RSNA Bone Age | NAtS | 0.0 | 83.8±8.7 | 99.9±0.0 | 99.1±1.1 | 99.0±2.0 | 80.2±7.2 | 99.9±0.0 |
| | | 0.2 | 67.2±3.4 | 88.1±5.7 | 89.2±4.0 | 64.0±2.5 | 65.7±5.2 | 96.1±2.4 |
| | | 0.4 | 54.7±1.7 | 62.2±3.9 | 73.1±5.2 | 46.4±4.8 | 53.9±1.7 | 84.9±2.7 |
| | | 0.6 | 52.0±1.3 | 52.1±2.6 | 60.3±4.8 | 42.3±2.9 | 51.5±1.6 | 68.9±1.5 |
| | | 0.8 | 50.3±0.9 | 48.9±1.5 | 52.3±1.4 | 40.8±1.6 | 50.0±1.1 | 54.3±1.3 |
| | ID | 1.0 | – | – | – | – | – | – |
| | NAtS | 1.2 | 51.6±1.0 | 54.7±1.5 | 56.5±1.7 | 42.5±2.1 | 51.6±1.1 | 53.9±2.4 |
| | | 1.4 | 55.0±1.5 | 64.4±2.9 | 65.8±3.5 | 47.3±2.4 | 55.1±1.4 | 62.9±3.8 |
| | | 1.6 | 58.9±2.8 | 73.2±3.6 | 73.8±4.3 | 54.0±2.9 | 58.9±2.2 | 72.4±4.9 |
| | | 1.8 | 61.3±3.2 | 81.0±3.9 | 79.4±5.5 | 58.1±3.2 | 60.9±2.2 | 78.8±5.4 |
| | | 2.0 | 61.9±3.5 | 87.2±3.6 | 83.2±6.0 | 60.1±4.7 | 61.3±2.6 | 83.6±5.0 |
| | | 2.5 | 63.9±3.5 | 94.7±2.2 | 89.9±5.6 | 66.7±6.1 | 63.1±3.0 | 91.6±3.6 |
| | | 3.0 | 69.9±3.0 | 97.9±1.0 | 93.8±4.3 | 74.6±5.5 | 69.1±2.4 | 95.0±2.5 |
| | | 3.5 | 73.8±3.2 | 98.9±0.5 | 95.4±3.8 | 79.0±5.9 | 72.9±2.0 | 96.5±2.1 |
| | | 4.0 | 76.6±4.0 | 99.3±0.4 | 96.5±3.1 | 83.7±6.5 | 75.4±2.1 | 97.2±1.8 |
| | | 4.5 | 78.2±4.7 | 99.4±0.4 | 97.3±2.4 | 86.9±5.9 | 76.5±3.5 | 97.6±1.6 |

Table 10: NAtS detection performance on three NAtS datasets measured by AUPR-Out.

# G  ADDITIONAL PCA ANALYSIS

In this section, we provide the additional PCA visualization results. Figure 8 presents PCA plots applied on the penultimate features obtained from the model trained with our loss. We observe that the shifted samples in the image domain move in between the two ID classes, and the majority of the NAtS samples do not overlap with the ID clusters. This helps in effectively detecting the shifted samples as NAtS.

Furthermore, we present the PCA visualization on a model trained with a cross-entropy loss in Figure 9. We categorize NAtS samples based on their grouth-truth labels as opposed to Figure 3 in which we do not segregate NAtS samples based on their ground truth labels. In the main paper, we intentionally do not categorize (and colorize) the NAtS samples because the ground truth class labels are not available at the inference time. In an ideal situation, the class labels should be predicted with the trained model only if the samples are not classified as NAtS, otherwise, they should be rejected for human assessment. Note that the PCA visualization cannot be used to infer the real increase in the distance between two classes since PCA compresses the original feature space into 2 dimensions.

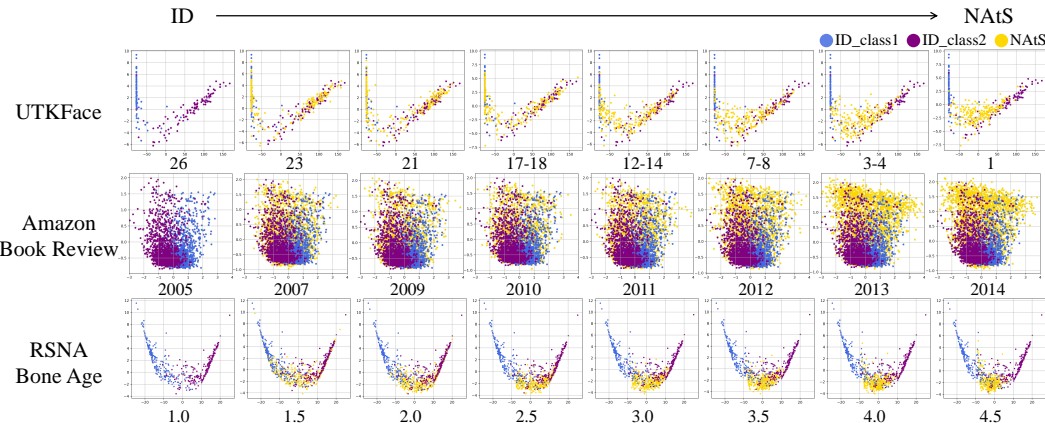

Figure 8: PCA visualization to demonstrate the movement of NAtS samples after training models with our proposed loss. We vary the age, year and brightness in UTKFace, Amazon Review and RSNA Bone Age dataset respectively.

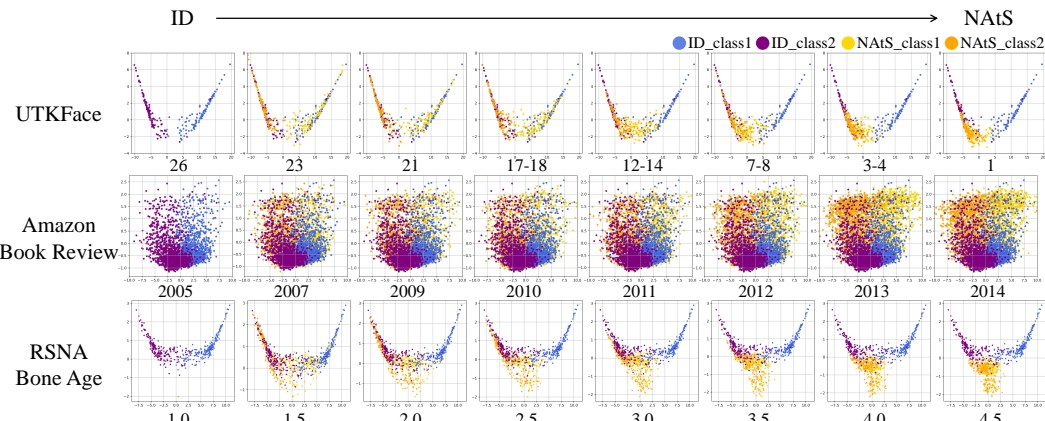

Figure 9: PCA visualization to demonstrate the movement of NAtS samples as we vary the age, year and brightness in UTKFace, Amazon Review and RSNA Bone Age dataset respectively.

# H    QUANTITATIVE RESULTS

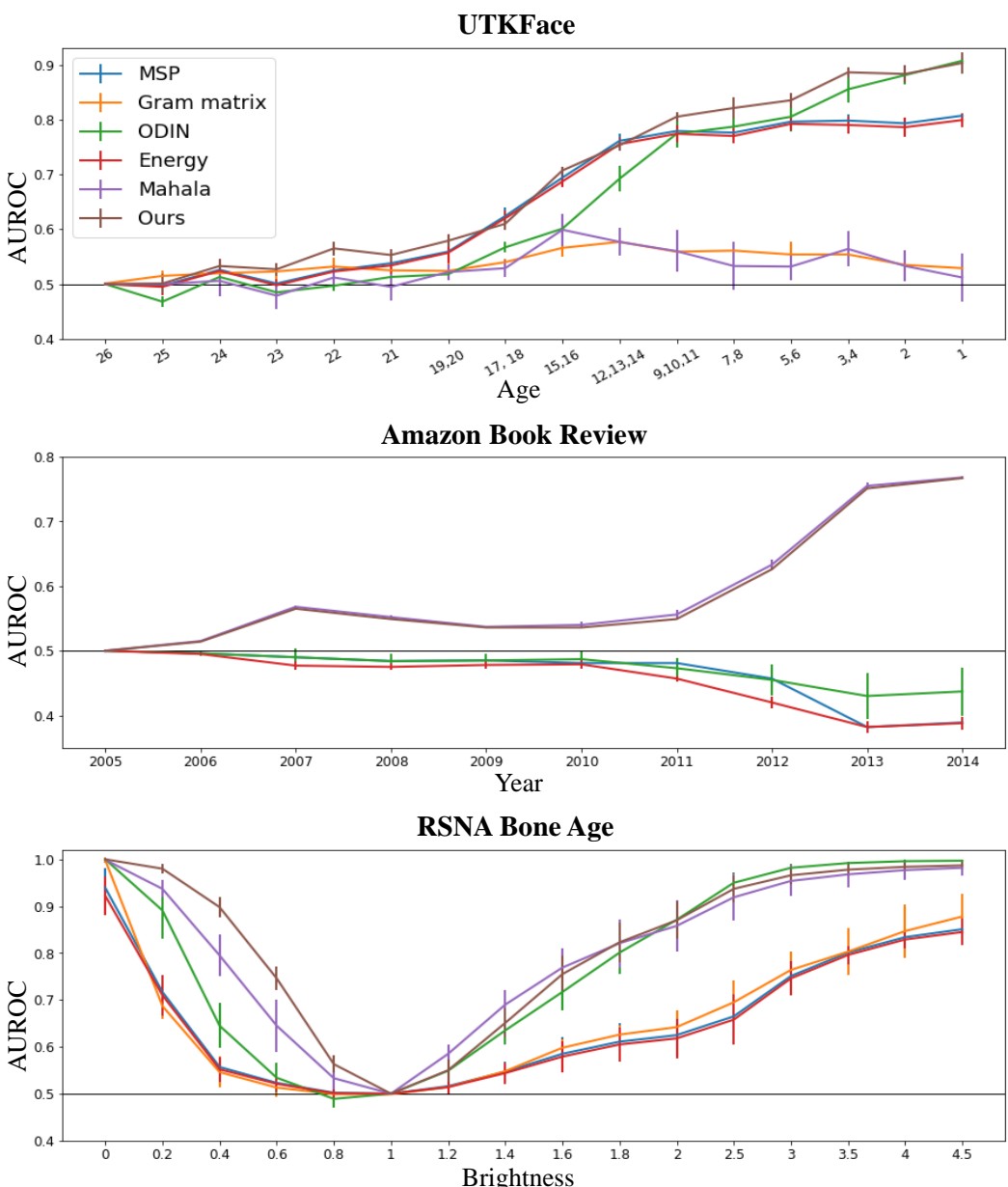

Figure 10: NAtS detection performance on distributional shifts in three datasets measured by AU-ROC.

# I EXPERIMENTS ON A MULTI-CLASS SYNTHETIC DATASET

We conduct additional experiments on a multi-class classification dataset to show that the three NAtS categories can exist in different tasks and datasets. In order to determine the NAtS category to which a shifted sample belongs, we need to train the model for a task on a dataset, evaluate the performance of existing OOD detection methods, and analyze the location of samples in feature space. However, we would have to conduct a series of experiments on numerous tasks and datasets until we encounter samples from each of the three NAtS categories. (There is no way to know the NAtS category of a given dataset until we run experiments on it.) To this end, we synthesized 2D multi-class dataset with 8 classes and conducted experiments on it.

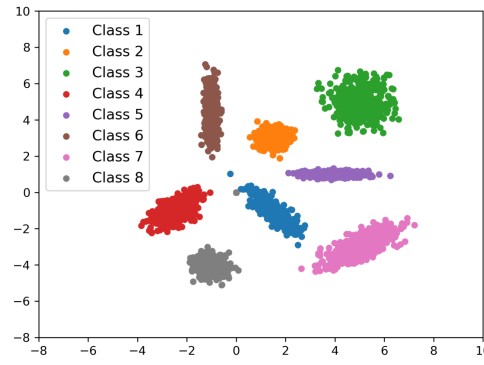

Figure 11: Visualization of synthetic dataset with 8 distinct classes. Each class label is defined by using the first two dimensions of each sample.

We synthesized the 2-dimensional vectors for training dataset with 8 classes, and those samples are visualized as red points in the 2D plane in Figure 12. In particular, we create samples in each class by randomly sampling from a gaussian distribution with a random mean and variance. Deep neural networks are primarily useful in complex datasets with high dimensional inputs. Therefore, for training, we increase the input dimensions by adding 8-dimensional vector which are filled with the Gaussian noise which have mean of zero and variation of 0.1. The network would focus on the 2 important input features which corresponds x and y coordinates to classify the classes and ignore others.

We train the feed-forward classifier using the standard cross-entropy loss and our modified loss. The classifier comprises 9 linear layers, and each hidden layer consists of 10 units, and the output layer consists of 8 units (one for each class). We then evaluate the performance of MSP, ODIN, Mahalanobis under NAtS for model trained using CE loss in Figure 12 (a), (b), and (c) respectively. **Purple Stars**, **Gray Diamonds** and **Orange Triangles** indicate samples from NAtS categories 1, 2 and 3 respectively. We observe that MSP and ODIN detects samples from NAtS category 1 and 3 but fails to detect from NAtS category 3. Mahalanobis detects samples from NAtS category 2 (**Gray Diamonds**) and 3 (**Orange Triangles**), but fails to detect samples from NAtS category 1 (**Purple Stars**). We then evaluate the performance of Mahalanobis under NAtS for model trained using our proposed loss in Figure 12 (d). We observe that Mahalanobis (on model trained with our loss) effectively detects samples from all NAtS categories. This demonstrates that our proposed training objective makes the Mahalanobis detector a robust NAtS detection method for all NAtS categories.

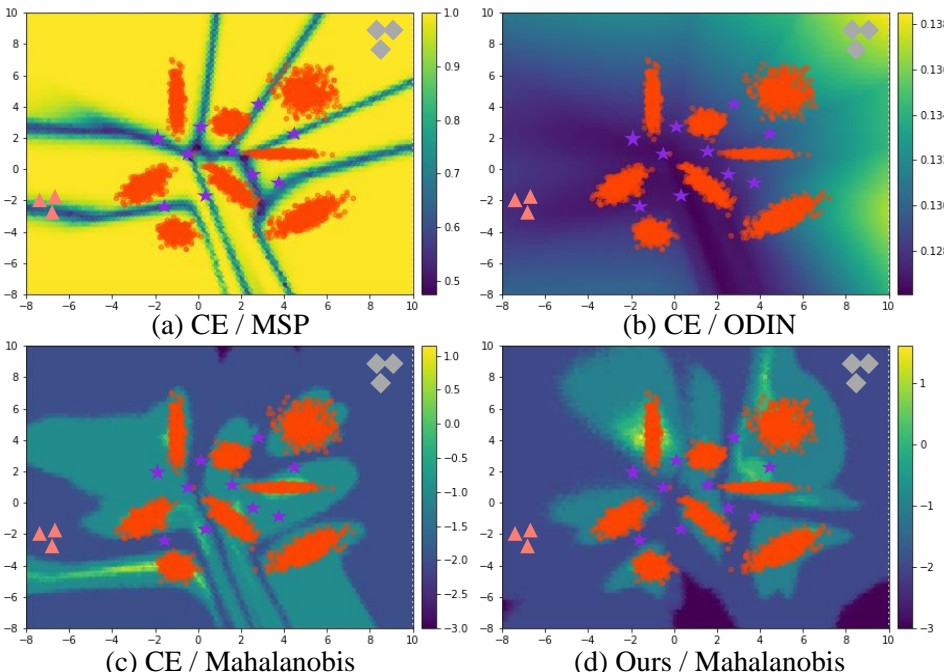

(a) CE / MSP

(b) CE / ODIN

(c) CE / Mahalanobis

(d) Ours / Mahalanobis

Figure 12: ID score landscape (brighter region means higher ID score) of the existing OOD detection methods in multi-class classification task. We use a synthetic 2D dataset with 8 distinct classes. The **Red points** represent the ID samples; **Purple Stars**, **Gray Diamonds** and **Orange Triangles** indicate samples from different NAtS categories. A sample is regarded as NAtS when it has a low ID score.

