# OpenReview forum: "Natural Attribute-based Shift Detection"
_ICLR.cc/2022/Conference — ICLR 2022 Submitted_

### Official Review · Reviewer_KqWM · 2021-11-02

**Correctness:** 4
**Technical Novelty And Significance:** 3
**Empirical Novelty And Significance:** 3
**Recommendation:** 8
**Confidence:** 3

**Main Review:**

The strengths of the paper are the following:
1. Present and analyze a new problem that OOD detection will fail to detection such NAS samples. Then categorize those samples into three categories.
2. Extensive experiments to demonstrate the ability of OOD detection methods to detect NAS.
3. New loss functions to improve classifiers so that OOD detection method, i.e. Mahalanobis, can detect NAS samples better.

The weaknesses of the paper are the following:
1. There is inconsistent in the setting of adjusting natural attributes between image, text and medical datasets. NAS samples in image and text datasets are from the real images while NAS samples in medical dataset are from synthetic images, i.e. changing brightness factor.
2. The proposed loss was used with Mahalanobis detector. How about the proposed loss with confidence-based OOD detectors?

**Summary Of The Paper:**

The paper defines a task called Natural Attribute-based Shift detection (NAS detection) to identify shifted samples on some natural attributes, e.g. age, time and lighting.
The authors create three benchmark dataset from existing datasets in three domains, i.e. vision, text and medical.
Their contributions are in three aspects: evaluation and analysis of out-of-distribution detection (OOD) methods on those NAS benchmark datasets, classify NAS samples as three different categories and present a modification to training objective of deep classifiers to improve an OOD detection method.

**Summary Of The Review:**

The paper is well written with extensive experiments. I think if the author could address my concern in the weakness listed above, the paper is good enough to publish in ICLR.

---

> ### Author Response · Authors · 2021-11-18
> **Response to Reviewer KqWM (R3)**
>
> **#1 There is inconsistent in the setting of adjusting natural attributes between image, text and medical datasets. NAtS samples in image and text datasets are from the real images while NAtS samples in medical datasets are from synthetic images, i.e. changing brightness factor.**
>
> Although NAtS dataset in the medical domain are synthesized by adjusting the brightness, they still reflect the probable scenarios in the real-world medical applications.
> In real-world medical applications, test data distribution is often shifted by some natural attributes such as brightness and contrast due to the various factors, e.g., incorrectly handling the imaging device, inconsistent lighting environment, variance across devices.
> Beede et al. surveyed the user experiences across some clinics in Thailand where the deep learning system for the detection of diabetic eye disease is deployed [1].
> Since the deep learning system is trained on clean and high-quality images, users should give such high-quality images as an input to obtain reliable answers from the system. In the research, out of 1838 images, 393 images did not meet the system’s high standards for reliable grading. Also, they report that such low-quality images having different conditions than the training dataset were obtained by ageing cameras or inexperienced environment settings (e.g., taking a photo in non-darkened space).
> Therefore, we believe our NAtS datasets in the medical domain reflect the probable scenario in real deployment even if they are synthesized.
> Further, since collecting dataset requires significant human and time resources in the medical domain, synthesizing other probable NAtS datasets by adjusting attributes and analyzing the dataset would be a promising future direction in the medical domain.
>
> [1] Beede et al., “A Human-Centered Evaluation of a Deep Learning System Deployed in Clinics for the Detection of Diabetic Retinopathy”, CHI, 2020
>
> **#2 The proposed loss was used with Mahalanobis detector. How about the proposed loss with confidence-based OOD detectors?**
>
> Thank you for the interesting question. Our training loss is proposed to improve the performance of a distance-based OOD detection method by increasing the distance between the in-distribution classes, therefore we expect that it would improve a distance-based OOD detection method and make it a robust NAtS detector.
>
> On the other hand, confidence-based OOD detection methods detect samples located near the decision boundary. Since our proposed training loss does not guarantee that NAtS samples will be located close to the decision boundary, it is difficult to expect that the performance of confidence-based OOD detection methods will improve.

---

> ### Author Response · Authors · 2021-12-03
> **Gentle reminder**
>
> Dear Reviewer KqWM,
>
> We thank you again for taking your precious time to review our paper and for providing us with your valuable and constructive feedback. We would really appreciate a reply as to whether our response and clarifications have addressed the concerns raised in your review. Please let us know if there is any further concern.
>
> We again thank you for your time and consideration!
>
> Best Regards,
> Authors

---

### Official Review · Reviewer_czxP · 2021-11-03

**Correctness:** 3
**Technical Novelty And Significance:** 4
**Empirical Novelty And Significance:** 3
**Recommendation:** 5
**Confidence:** 4

**Main Review:**

strengths
* The idea is novel.
* The writing is easy to understand.
* This work improves the generalizability of the distance-based Mahalanobis method for NAS under different tasks.

weaknesses
* Some assumptions lack theoretical support. Why do the natural properties of age and image brightness make unreliable predictions **more severe**?
* The experiment setting is weak. It is not convincing to support the claim with just three OOD methods and three binary classification tasks.
* The acronym for the new task should be distinguished from the abbreviation for the Neural Architecture Search.
* Table 2 is too far from the paragraph where it was first cited.
* The formulation in Section 4.2 lacks explanation, making it hard to be reproduced.
* To make sure the NAS could be detected, the model must be trained by adding extra loss, making the proposed method inefficient. Though the performance has not shown significant degrades, the time cost can not be ignored.
* The study on different NAS situations is not thorough enough.
* The authors should also visualize feature distribution after improving the Mahalanobis method using PCA.

**Summary Of The Paper:**

The authors propose a new task named NAS (natural attribute-based shift), which is a sub-task of OOD (out of distribution). Three datasets are designed from existing datasets for the NAS experiment. By visualizing the distribution of category features produced by the model under different datasets using PCA, the authors find three categories for NAS distribution. Total three OOD methods are tested on NAS datasets and show an inconsistency in their performance. Finally, the authors propose to use distance loss and entropy loss to retrain the models to improve the distance-based OOD methods.


**Summary Of The Review:**

Overall, I like the idea of this article, the proposed new task is valid. However, there are some weaknesses, especially the experiments that are not comprehensive and convincing enough.

---

> ### Author Response · Authors · 2021-11-18
> **Response to Reviewer czxP (R2) (1/2)**
>
> **#1 Some assumptions lack theoretical support. Why do the natural properties of age and image brightness make unreliable predictions more severe?**
>
> We apologize for confusing the reviewer with a rather unclear statement. What we intended to describe is that, when the test distribution is clearly different from the training distribution (e.g. training an image classifier on MNIST and testing it on CIFAR-10), it is relatively easier to recognize that your classifier is making unreliable predictions (i.e. you can’t trust your model) via some OOD detection method. But when the test distribution is gradually shifting due to a natural attribute, it is relatively harder to decide when to stop trusting your model, because it seems to be working just fine when the amount of shift is negligible. We revised the said sentence to more clearly convey our intention.
>
>
> **#2 The experiment setting is weak. It is not convincing to support the claim with just three OOD methods and three binary classification tasks.**
>
> In this work, we observe the inconsistency of NAtS detection performance of confidence- and distance-based OOD detection methods.
> Although we mainly use three OOD detection methods for our initial analysis, we confirm that the
> same problem (inconsistent detection performance depends on the NAtS categories) also exists in the other OOD detection methods (OOD detection methods which utilize Gram matrix and energy score) in Table 3.
> We believe these results show that our claim is not constrained on the three OOD detection methods, but it holds true for all confidence- and distance-based OOD detection methods.
>
> Also, the three binary classification tasks and its corresponding NAtS scenarios represent all three NAtS categories with datasets from multiple domains.
> Thus, we believe our suggested three classification tasks are sufficient to support our claim.
>
> However, to address the concern and further bolster our method's effectiveness, we conduct experiments on a multi-class classification dataset to show that the three NAtS categories can exist in different tasks and datasets.
> To determine the NAtS category to which a shifted sample belongs, we need to train the model for a task on a dataset, evaluate the performance of existing OOD detection methods, and analyze the location of samples in feature space. However, we would have to conduct a series of experiments on numerous tasks and datasets until we encounter samples from each of the three NAtS categories. (There is no way to know the NAtS category of a given dataset until we run experiments on it)
> To this end, we synthesized 2D multi-class dataset with 8 classes and conducted experiments on it. We provide details of the experiments and results in Section I of the Appendix.
>
>
> **#3 The acronym for the new task should be distinguished from the abbreviation for the Neural Architecture Search.**
>
> We are thankful to reviewer 1 and 2  for pointing this out. We decided to change the abbreviation of Natural Attribute-based Shift as NAtS to easily distinguish it from the abbreviation of Neural Architecture Search.
>
> **#4 Table 2 is too far from the paragraph where it was first cited.**
>
> Thank you for your detailed comment.
> We changed the location of Table 2 close to Section 4.2 where it was first cited. Table 1 and Table 2 in the modified version corresponds to Table 2 and 1, respectively, in the previous version.
>
> **#5 The formulation in Section 4.2 lacks explanation, making it hard to be reproduced.**
>
> We describe the implementation and training details, along with information on computation resources in Section C of the Appendix.
> We will clarify that such an explanation is provided in the Appendix at the end of the section.

---

> > ### Author Response · Authors · 2021-11-18
> > **Response to Reviewer czxP (R2) (2/2)**
> >
> >
> > **#6 To make sure the NAtS could be detected, the model must be trained by adding extra loss, making the proposed method inefficient. Though the performance has not shown significant degrades, the time cost can not be ignored.**
> >
> > Thank you for raising this interesting question.
> > Indeed we trained the model with additional loss terms to ensure that NAtS samples are effectively detected. In the table below, we present time taken per epoch on training the models with and without the proposed loss terms for the three classification tasks.
> >
> >
> >
> > Time per epoch (in seconds)
> >
> > |         | Time per epoch (in seconds) |        |
> > |---------|-----------------------------|--------|
> > |         |              CE             |  Ours  |
> > | Image   |                       2.898 |  3.088 |
> > | Text    |                       18.35 |  18.92 |
> > | Medical |                      16.079 | 16.232 |
> >
> >
> >
> > From the table, we observe that there is no significant difference in training the model using our proposed loss terms in the vision, text and medical domain. Therefore, for all our experiments, we find that our proposed loss term does not have a significant effect on efficiency.
> >
> > However, if we consider large datasets (e.g., imagenet) to test the scalability of our method, we agree that additional loss terms (especially, distance loss) may be computationally expensive. As addressed to R1, in such cases, we can group the output classes based on some prior knowledge (as was done in MoS [1]), or a clustering method to reduce the number of terms. Testing our method on large datasets and improving its scalability would be a promising future direction.
> >
> > [1] Huang et al., MOS: Towards Scaling Out-of-distribution Detection for Large Semantic Space, CVPR, 2021
> >
> > **#7 The study on different NAtS situations is not thorough enough.**
> >
> > In this work, we divided the whole problem domain based on two criteria: 1. whether the dataset is near the decision boundary or not; 2. whether they are close or far from the in-distribution dataset. Thus, there are four possible categories. Since a dataset (i.e., samples) far from the decision boundary and overlapping with the in-distribution dataset is not a distributionally shifted dataset, our work focuses on the remaining three categories that cover all possible scenarios of NAtS.
> > Therefore, we believe that we have included all possible categories for NAtS samples under the two criteria. Also, the degree of distance from ID was addressed by adjusting the semantic attributes (i.e., age/time/brightness) with various levels.
> >
> > **#8 The authors should also visualize feature distribution after improving the Mahalanobis method using PCA.**
> >
> > Thank you for this interesting suggestion. We add the visualizations of the feature representations of ID and NAtS samples obtained after training the model with the proposed loss terms in Section G of the Appendix. We observe that without our loss, such samples are not detected as NAtS because the classes are not separated enough and the majority of the shifted samples overlap with the ID samples. However, when the model is trained with our loss, the inter-class distance increases which provides more space for shifted sampes to move around.
> > From Figure 8, we observe that the shifted samples in the image domain move in between the two ID classes and majority of the NAtS samples do not overlap with the ID clusters and therefore they can be effectively detected as NAtS samples.
> >
> > Note that the PCA visualization cannot be used to infer the real increase in the distance between two classes since PCA compresses the original feature space into 2 dimensions. Instead, we can guarantee the increase in the distance between classes after using our proposed loss terms by empirically calculating the distance between mean features of the classes.

---

> ### Author Response · Authors · 2021-12-03
> **Gentle reminder**
>
> Dear Reviewer czxP,
>
> We thank you again for taking your precious time to review our paper and for providing us with your valuable and constructive feedback. We would really appreciate a reply as to whether our response and clarifications have addressed the concerns raised in your review. Please let us know if there is any further concern.
>
> We again thank you for your time and consideration!
>
> Best Regards,
> Authors

---

### Official Review · Reviewer_bsNT · 2021-11-06

**Correctness:** 4
**Technical Novelty And Significance:** 3
**Empirical Novelty And Significance:** 4
**Recommendation:** 6
**Confidence:** 4

**Main Review:**

STRENGTHS
1. I like the the idea of Natural Attributed-based Shift as an important subset of Out-of-Distribution problems we can face in DL systems. I think it is sufficiently different from typical OOD problems studied in literature (different label space - CIFAR vs ImageNet or different support - images vs sketches). I don't completely buy the argument that one is harder or easier compared to the other (paragraph 2 of Section 1) but I think a systematic study of this problem is important.
2. The three proposed datasets of: (1) Different grouping of UTKFace, (2) Different grouping of Amazon Review and (3) RSNA Bone Age dataset at different brightness settings, are good examples of interest that capture different domains (image, test and medical diagnosis).
3. Identification of 3 types of NAS scenarios based on whether the dataset is close to decision boundary (or not) and if it is close to the in-distribution data (or not) is helpful especially since it identifies which OOD methods are expected to work for a particular scenario.
4. The additional loss terms help enforce the constraints identified in the paper to enable better detection of OOD samples for the Mahanalobis model. The paper empirically demonstrates that they get comparable performance to the original model (even with the new terms) and are able to show good performance in the UTKFace dataset where the Mahanalobis model was not doing well earlier.

MAJOR COMMENTS:
1. I think NAS is studied implicitly in many circumstances related to pose for example, how to handle side faces for a DL system trained on frontal faces or similar pose problems in 3/6-dof object pose or medical image pose where registration followed by applying the system is an imperfect solution. Studying the NAS problem explicitly is expected to provide better solutions but I don't see any discussion of such a dataset or problem in the paper. I'd like the authors to comment on if they consider such a problem to be a NAS problem and if their proposed solutions apply to this NAS setting? One dataset of interest might be the [3D Object Dataset](https://ieeexplore.ieee.org/abstract/document/4408987) where a certain range of viewpoints is D_I and the others are D_S.
2. Table 3 is very dense and not easy to parse. I strongly recommend converting it to the same format as Figure 2 for (a) easy comparison with Figure 2 to measure the impact of additional loss terms and (b) Visual Interpretation of the results.
3. It is unclear how this method scales to say the ImageNet dataset with 1000 categories and a ResNet-101 model with penultimate layer of size 2048. The additional terms might be too expensive at such scale for training. Any thoughts/comments?

MINOR COMMENTS:
1. NAS in Deep Learning Literature has been commonly used for Neural Architecture Search. I would recommend not using NAS for Natural Attribute-based Shift as well.
2. In Introduction: "benchmark datasets in vision, language, and medical for NAS detection". The word "medical" is incomplete here. I think you mean medical domain?
3. In Section 4.2, paragraph "Experimental Settings", why use ResNet-18 instead of the more common ResNet-50?
4. In Figure 2, add a note saying Higher is better? Also, for all the figures (a), (b) and (c), the AUROC is at 0.5 for D_I. I could not find any text explaining if some normalization was done for the results or if these are raw AUROC values.
5. In Section 5.1, why use PCA instead of something like t-SNE?
6. In Figure 3, can you change it to show NAS_class1 and NAS_class2 as well please? That will be more informative compared to just NAS shown right now.
7. Why not include Gram matrix and Energy score in Figure 2 analysis?

**Summary Of The Paper:**

The paper identifies the problem of a Natural Attributed-based Shift (NAS) in the data distribution affecting Deep Learning (DL) systems designed for the task of interest. An example may be a DL system for X-ray based diagnosis being able to handle a different brightness that it was trained for. The paper argues that it is similar to Out-of-Distribution (OOD) data but instead of the difference being in the label space (CIFAR vs SVHN) the label space is similar but the test data distribution is shifted due to a systemic shift in an important data attribute (brightness of image in the X-ray example above). The paper claims the following contributions:
- A new task of NAS detection and datasets to study it.
- A comprehensive evaluation of OOD methods on the NAS dataset and demonstration that none of them perform consistently across all NAS scenarios.
- A novel analysis based on the location of shifted samples in the feature space and performance of existing OOD methods.
- Demonstrate that a simple modification to training objective for deep classifiers enables consistent OOD performance on all NAS scenarios.


**Summary Of The Review:**

Correctness: 4
Technical Novelty and Significance: 3 (I am debating between 3 and 4 for the final review. I think the NAS problem is clearly identified and the datasets curated for it are novel but the problem itself, of OOD, is not novel. There are more empirical insights than theoretical ones for the paper. I might change this rating based on the other reviews / discussions).
Empirical Novelty and Significance: 4 (I am debating between 3 and 4 for the final review. I think the total sum of contributions: detailed analysis, identification of the NAS scenarios, and additional loss terms to Mahanalobis detector make it novel & significant even if some parts may not be. I might change this rating based on the other reviews / discussions).

Overall Rating: Weak Accept
Given the strengths and weaknesses of the paper mentioned above, I recommend an Weak Accept for the paper. It has good contributions with the NAS problem & relevant datasets to test it. It also presents a good analysis of existing OOD methods, their flaws and a potential improvement in one (for the Mahanalobis detector). My main concerns are about the Pose component of NAS and scalability of the proposed modification (new loss terms) which if addressed/responded to in the rebuttal discussion will help me change to an Accept.

---

> ### Author Response · Authors · 2021-11-18
> **Response to Reviewer bsNT (R1) (1/2)**
>
>  **#1 I think NAS is studied implicitly in many circumstances related to pose for example, how to handle side faces for a DL system trained on frontal faces or similar pose problems in 3/6-dof object pose or medical image pose where registration followed by applying the system is an imperfect solution. Studying the NAS problem explicitly is expected to provide better solutions but I don't see any discussion of such a dataset or problem in the paper. I'd like the authors to comment on if they consider such a problem to be a NAS problem and if their proposed solutions apply to this NAS setting? One dataset of interest might be the 3D Object Dataset where a certain range of viewpoints is D_I and the others are D_S.**
>
> Thank you for sharing this interesting direction. We believe that we can consider this as a NAtS problem where samples with viewpoints from D_I can be considered as ID samples and samples with viewpoints from D_S can be considered NAtS samples, the viewpoint being the natural attribute. It seems to be a promising future direction to evaluate our task on 3D datasets and solve important problems, including, detection of  samples shifted by natural attributes such as viewpoints.
>
> **#2 Table 3 is very dense and not easy to parse. I strongly recommend converting it to the same format as Figure 2 for (a) easy comparison with Figure 2 to measure the impact of additional loss terms and (b) Visual Interpretation of the results.**
>
> As R1 suggested, we convert Table 3 to the same format as Figure 2 and include the plot in the Section H of the Appendix. However, we found overlapping line plots for some OOD detection methods (especially between MSP and Energy-based) at some segments.  Also, this figure was very similar with Figure 2, which can be considered as redundant information because the results of MSP, ODIN, Mahalanobis are duplicated. Therefore, after giving a lot of thought to it, we decided to present the plot version of Table 3 in Section H of the Appendix. If you still believe that presenting these results in figure instead of table will be more effective, we will include the figure in the main paper of the next revision.
>
>
> **#3 It is unclear how this method scales to say the ImageNet dataset with 1000 categories and a ResNet-101 model with penultimate layer of size 2048. The additional terms might be too expensive at such scale for training. Any thoughts/comments?**
>
> Thank you for raising an interesting and important point. The distance loss in Eq.2, in its naive form, requires xC2 terms where x is the number of output classes. When the x is a moderate number (10-100), we believe adding xC2 terms in the objective function would not be a huge problem for modern computers. However, when x becomes as large as 1,000 it would be better to group the output classes based on some prior knowledge (as was done in MoS [1]), or a clustering method to reduce the number of terms. Testing our method and improving its scalability would be a promising future direction.
>
> [1] Huang et al., MOS: Towards Scaling Out-of-distribution Detection for Large Semantic Space, CVPR, 2021
>
> **#4 NAtS in Deep Learning Literature has been commonly used for Neural Architecture Search. I would recommend not using NAtS for Natural Attribute-based Shift as well.**
>
> As R1 and R2 suggested, we decided to change the abbreviation of Natural Attribute-based Shift as NAtS.
>
> **#5 In Introduction: "benchmark datasets in vision, language, and medical for NAtS detection". The word "medical" is incomplete here. I think you mean medical domain?**
>
> Thank you for the detailed review. In the revised version of our paper, we change the word as “medical domain” to clarify our intention.

---

> > ### Author Response · Authors · 2021-11-18
> > **Response to Reviewer bsNT (R1) (2/2)**
> >
> > **#6 In Section 4.2, paragraph "Experimental Settings", why use ResNet-18 instead of the more common ResNet-50?**
> >
> > Compared to the conventional setting of out-of-distribution detection tasks, our setting has a relatively small number of data samples and classes. Specifically, in our setting, the in-distribution dataset of the image domain consists of 1,282 facial images with 2 classes, and that of the medical domain has 3,000 images with 2 classes. In contrast, both CIFAR-10 and CIFAR-100, which are commonly used for in-distribution dataset in out-of-distribution detection tasks, consist of 60,000 samples with 10 classes. Therefore, in our experiments, the architecture of ResNet-18 was enough to train each model and obtain satisfactory classification accuracy (94.1% and 92.3% in image and medical domain, respectively)
> >
> > **#7 In Figure 2, add a note saying Higher is better? Also, for all the figures (a), (b) and (c), the AUROC is at 0.5 for D_I.  I could not find any text explaining if some normalization was done for the results or if these are raw AUROC values.**
> >
> > As R1 suggested, we add the detailed explanation about AUROC in the revised version of our paper. The higher AUROC indicates that more NAtS samples are distinguished from the ID samples by using the detection method. In Figure 2, we additionally calculate the AUROC values where there is no shift in the attribute to provide the reference point. Specifically, we duplicate the D_I dataset which is the ID dataset in the scenario and consider the copied dataset of D_I as NAtS data with a shift degree of zero. Since ID and NAtS data are the same, the two datasets cannot be distinguished. Therefore, the AUROC score would be 0.5 for D_I.
> >
> >
> > **#8 In Section 5.1, why use PCA instead of something like t-SNE?**
> >
> > In the paper, we used PCA since we wanted to analyze the different ID and NAtS samples based on their relative location in the feature space.
> > The idea of t-SNE is to place similar samples close to each other, often ignoring the global structure.
> > In other words, from t-SNE plots, it cannot be guaranteed that the relative distance between the samples in 2D reflects the true distance in the original latent space.
> > Therefore, we used PCA, which preserves the global structure and helps in analyzing the relative position of different samples in the feature space.
> >
> > **#9 In Figure 3, can you change it to show NAS_class1 and NAS_class2 as well please? That will be more informative compared to just NAS shown right now.**
> >
> > Thank you for raising this question. In Figure 3, we intentionally do not categorize (and colorize) the NAtS samples. At the inference time, the ground truth class labels are not available. In an ideal situation, the class labels should be predicted with the trained model only if the samples are not classified as NAtS, otherwise, they should be rejected for human assessment. However, upon your request, we provide another version of Figure 3 with NAtS samples colorized based on their ground-truth class labels in Section G of the Appendix.
> >
> > **#10 Why not include Gram matrix and Energy score in Figure 2 analysis?**
> >
> > In this paper, we probe the confidence- and distance-based OOD detection methods under three different NAtS scenarios. At the initial experiment, to keep the analysis succinct and to-the-point, three well-known and representative methods were mainly addressed to show there is an inconsistency on detection performance depending on the NAtS category (using more detection methods would have increased the volume of the initial sections while not bringing new insights to the table).
> >
> > After that, we show that other OOD detection methods utilizing the energy score and gram matrix have similar results with confidence- and distance-based OOD detection methods respectively. This demonstrates our claim is not constrained on just three detection methods, but it generally holds for other confidence- and distance-based methods. To develop our claim effectively with the succinctness of the analysis, we focus on the three representative OOD detection methods (MSP, ODIN, Mahalanobis) in the initial experiments including Figure 2.

---

> ### Author Response · Authors · 2021-12-03
> **Gentle reminder**
>
> Dear Reviewer bsNT,
>
> We thank you again for taking your precious time to review our paper and for providing us with your valuable and constructive feedback. We would really appreciate a reply as to whether our response and clarifications have addressed the concerns raised in your review. Please let us know if there is any further concern.
>
> We again thank you for your time and consideration!
>
> Best Regards,
> Authors

---

### Author Response · Authors · 2021-11-18
**General Response**

We would like to thank all reviewers for providing high quality reviews and constructive feedback. For convenience, we will denote Reviewer bsNT, Reviewer czxP, and Reviewer KqWM as R1, R2, and R3, respectively.

We were encouraged to see that reviewers found the problem of NAtS detection novel (R1, R2, R3) and clearly identified (R1); the detailed analysis of existing OOD methods interesting (R1, R3); additional loss terms to improve Mahalanobis detector novel (R1, R3),  significant (R1), and strength of our work (R2); the paper well written (R3) and easy to understand (R2); curated NAtS datasets novel (R1); experiments extensive (R3); and acknowledged the contributions of identification of NAtS categories (R1). We are pleased to see that the reviewers found the contributions of our work significant (R2).

We addressed reviewers’ concerns and suggestions in our reviewer-specific responses.
The valuable suggestions and reviews by the reviewers helped us improve the draft, and the modifications in the new version are summarized below:

* [R1, R2] Changed the abbreviation of Natural Attention-based Shift from **NAS** to **NAtS**.
* [R1] Converted the results in Table 3 to a plot of the form shown in Figure 2 and included the plot in Section H of the Appendix.
* [R1] Changed the word “medical” as “medical domain” in Introduction.
* [R1] Added comments in Section 4.2 to clarify the detailed information of the AUROC in Figure 2.
* [R1] Added Figure 9  in Section G of the Appendix to show the PCA visualization with the label information of NAtS.
* [R2] Improved some sentences in the Introduction to clearly convey our intention.
* [R2] Changed the location of Table 2 close to Section 4.2.
* [R2] Added a comment in Section 4.2 stating that Section C in Appendix provides implementation details.
* [R2] Added Figure 8 in Section G of the Appendix with PCA visualizations after training models with our proposed loss.
* [R2] Added the analysis of NAtS categories under the multi-class classification setting in Section I of the Appendix.

---

### Decision · Program_Chairs · 2022-01-20

**Decision:**

Reject

**Comment:**

This paper offers "natural attribute" methods for shift detection.  Several reviewers are positive, but reviewer czxP is the most authoritative in the eyes of the area chair.  In particular the AC is concerned that the task that this paper defines is artificial and not useful. Naturally occurring shifts are real, happen all the time and meaningfully affect model performance... Natural shifts should be defined over distributions not singular instances. The authors create artificial instantiations of natural shifts to illustrate a well known flaw in OOD detection algorithms.
To demonstrate the usefulness of this approach to "natural shifts" the authors should should show how this algorithm performs in settings like "WILDS"... where the shift are meaningful and not artificial, in the opinion of the AC.  The paper should cite and contrast WILDS (https://arxiv.org/abs/2012.07421) and Mandoline (http://proceedings.mlr.press/v139/chen21i.html) in a future revision.